

# A Sample Characterization Toolkit for Carbonate U-Pb Geochronology

E Troy Rasbury[1], Theodore M Present[2], Paul Northrup[1], Ryan V Tappero[3], Antonio Lanzirotti[4], Jennifer M Cole[5], Kathleen M Wooton[1], and Kevin Hatton[1]

[1]Department of Geosciences, FIRST, Stony Brook University, 100 Nichols Rd, Stony Brook, NY 11794, USA
[2]Division of Geological and Planetary Sciences, California Institute of Technology, 307 North Mudd Laboratory, Pasadena, CA 91125, USA
[3]NSLS-II, Brookhaven National Laboratory, Upton NY 11973, USA
[4]Center for Advanced Radiation Sources, Randall, Chicago, IL 60637
[5]Department of Science, West Los Angeles College, Culver City, CA 90230, USA

**Correspondence:** E. Troy Rasbury (troy.rasbury@stonybrook.edu))

**Abstract.** Laser ablation U-Pb analyses of carbonate (LAcarb) samples has greatly expanded the potential for U-Pb dating to a variety of carbonate producing settings. Carbonates that were previously considered impossible to date using isotope dilution methods may preserve domains that are favorably interrogated when using spatially resolved laser ablation geochronology techniques. Work is ongoing to identify reference materials and to consider best practices for LAcarb. In this study we apply 5 standard and emerging characterization toolsets on three natural samples with the dual goal of enhancing the study of carbonates and in establishing a new set of precisely characterized natural standards for LAcarb studies. We start with the existing carbonate reference material WC-1 from the Permian Reef Complex of Texas, building on the published description to offer a deeper look at U and fluids. We consider a tufa sample from the Miocene Barstow Formation of the Mojave Block, California, as a possible secondary calcite reference material due to its well-behaved U/Pb systematics. There are currently no natural 10 dolomite standards. We present an unusual dolomite sample with very well-behaved U-Pb systematics from the Miocene of the Turkana Basin of Kenya as a possible dolomite reference material for LAcarb dating. In addition to using XRF mapping and spectroscopy to better understand U in these natural samples, we have analyzed multiple aliquots of each of them for $^{87}Sr/^{86}Sr$. The Sr isotope compositions are reasonably homogeneous in all three samples, so that these could be used as Sr isotope standards as well. This combination could streamline split stream analyses of $^{87}Sr/^{86}Sr$ and U/Pb geochronology.

## 1 Introduction

Carbonates are ubiquitous in the Earth's crust and form from a variety of fluids, often with characteristic isotopic and element ratios (Fig. 1). Carbonate producing fluids range from oxygenated to reducing, from low salinity to high salinity and from low T to high T. Fluids evolve as they interact with rocks along their flow paths, adding complexity, but also providing an opportunity to consider these processes in more detail. Carbonates are highly susceptible to diagenesis, particularly when the 20 fluid chemistry and temperature changes, bringing them into and out of the carbonate stability fields. Mixing of fluids can be particularly corrosive or can be responsible for mineral precipitation. Introduction of fluids with different chemistries through



uplift or burial can destroy the original carbonate prior to precipitation of a new carbonate or can alter the carbonates visibly or at a microscopic scale. Therefore, an important part of any investigation of carbonates should involve establishing field and petrographic relationships. We assume if you are working with carbonates, that your toolkit already involves establishing

field relationships and careful petrography. You have a context that involves knowing the depositional and diagenetic histories, and our focus here is more on elucidating of the geochemsitry as it relates to the potential for incorporation of U and Pb in carbonates.

It is widely recognized that carbonates are a proxy for the fluids that formed them. Published studies demonstrate the need for careful characterization of carbonates in order to obtain meaningful results, and this same attention to detail is needed to

understand U/Pb results. Carbonates have element ratios that are controlled by the composition of the fluid and the Kd of the element for the crystal lattice (Banner and Hanson, 1990). Special circumstances are required to have elevated U in calcite because the Kd is less than 0.05 (Reeder et al., 2001; Drysdale et al., 2019; Weremeichik et al., 2017). However, aragonite, which is metastable, does not exclude U (Reeder et al., 2001) and Kelly et al. (2003) suggested that high U contents in a speleothem calcite (which we use as a U(VI) standard), might have resulted from neomorphism of original aragonite. No

experimental studies for U incorporation have been reported for dolomite, but it is unlikely to have a higher affinity for U than calcite as they have the similar structures, and there is more of a size mismatch with Mg than Ca. Using modern settings and a laboratory understanding of trace element incorporation, we know that U incorporation is higher in aragonite > high Mg calcite> calcite (Chung and Swart, 1990; Reeder et al., 2001). Seawater has elevated U/Pb, but carbonates forming from seawater have much lower U/Pb, demonstrating that the structure has a much higher affinity for Pb than U (Shen and Boyle,

1988), at least at seawater composition. A missing component here is that no lab experiments have studied U(IV) incorporation in carbonates. There are several published examples of natural carbonates with reduced U (Sturchio, 1998; Cole et al., 2004) and we present one more in this contribution. Clearly more work is needed to understand how reduced U is available to go into carbonates. Experimental work at elevated T has shown that U(IV)Cl is soluble in acid solutions (Timofeev et al., 2018), and high T experiments of U in hydrocarbons indicate that economic concentrations can be produced via transport in hydrocarbons

(Migdisov et al., 2017), but how would these processes result in incorporation in carbonates?

Cement stratigraphy is a powerful approach to establish the relative age relationships of carbonate phases, or paragenesis across a basin. This approach uses relative relationships of cements from the thin section scale and correlation of these throughout a basin or region (Meyers, 1991, 1974). This field to petrographic approach has been used to select samples for geochemistry, providing fundamental insights into the fluid evolution of basins and degree of rock/water interaction e.g. (Ban-

ner et al., 1988), as well as establishing normal trends in Mn/Fe with burial (Barnaby and Rimstidt, 1989). U-Pb dating in this larger context places age constraints on the fluids which can then be tied to testable hypothesis about the origin and changes in the fluids (Lawson et al., 2018; Quade et al., 2017; Parrish et al., 2019; Pisapia et al., 2018; Woodhead et al., 2010; Winter and Johnson, 1995; Hoff et al., 1995; Cole et al., 2005; Polyak et al., 2008, 2013; Luczaj and Goldstein, 2000; Brannon et al., 1996; Godeau et al., 2018; Rasbury et al., 2004, 1997; Pagel et al., 2018; Roberts et al., 2020).

Laser ablation mapping (Woodhead et al., 2010, 2007; Piccione et al., 2019; Drost et al., 2018; Roberts et al., 2020), synchrotron XRF mapping (Cole et al., 2004; Piccione et al., 2019; Frisia et al., 2008; Vanghi et al., 2019), PIXE (Ortega et al.,



2005, 2003) and μXRF (de Winter et al., 2017) are all ways to take the physical observations of different phases to a new level with in situ chemistry, that maps U in particular, along with other elements that provide details of the fluid chemistry. For the purpose of U-Pb dating, knowledge of mineral lattice location of U and it's oxidation state and speciation provide a fundamen-

60 tal framework for interpreting ages as well as revealing details of the fluid chemistry (Kelly et al., 2006, 2003; Sturchio, 1998; Cole et al., 2004). Given that the major element composition of seawater has changed appreciably through time (Hardie, 1996; Horita et al., 2002), that meteoric fluid compositions are controlled by water rock interaction (Chung and Swart, 1990) and that deep brines have evolved since deposition in basins (Musgrove and Banner, 1993), we can imagine that there is no such thing as a typical fluid. A holistic approach to dating carbonates should involve an effort to see back to the fluid or fluids that

have been responsible for its formation and how they might have changed through the diagenetic history. This, particularly in the context of how favorable the U concentrations and U/Pb ratios are, will provide an important path towards understanding factors that create favorable carbonate samples for U-Pb dating. The purpose of this contribution is to highlight tools that are available for carbonate characterization relevant to U-Pb dating.

## 1.1 Mapping Uranium

While it is the U/Pb ratio that matters for dating, it is usually the U concentration that limits the potential for dating, so the focus here will be on tools that map the distribution of U on the hand specimen to thin section scale.

  Fission track mapping (Chung and Swart, 1990; Hoff et al., 1995; Rasbury et al., 2000; Cole et al., 2004; Becker et al., 2002; Wang et al., 1998; Luczaj and Goldstein, 2000; Amiel et al., 1973; Haglund et al., 1969; Gvirtzman et al., 1973) can help to map the distribution of U on the microscopic to hand specimen scales. This simply requires making a polished slab and placing

it on a detector and sending it to a nuclear reactor. Depending on the flux requested, it typically takes weeks for the samples to be returned due to harmful radioactivity produced in the reactor. Once back, the detector has to be etched to increase the size of the tracks for visibility and then a comparison to the rock can be made. Thin sections can be used, making it easier to relate the tracks to the exact phases. There is a modest cost for sending samples to a reactor, but this is an underutilized non-destructive approach that could be followed by LAcarb methods. For samples with high U (10's of ppm), modern high resolution scanners

can image the tracks. For lower concentration samples, high magnification microscope images can be mosaiced to provide a larger scale framework for the U distribution.

  Autoradiography (Cole et al., 2003; Pickering et al., 2011) offers the opportunity to examine radioactive elements on the hand specimen scale. Phosphor imaging plates are used in biomedical and biochemical studies to detect samples tagged with radionuclides. While the scanning devices themselves are somewhat costly, phosphor screens can be rented from imaging

centers or even purchased at lower cost, making this technique more broadly applicable. Polished hand samples along with a solid standard are left to activate the phosphor over time (days to weeks), ideally in a shielded environment. Following exposure, the screens are scanned to produce a digital autoradiograph that provides a map of relative differences in U and Th concentrations directly comparable to the hand specimen (Cole et al., 2003; Pickering et al., 2011). Cm-scale regions of higher U (and Th) can then be targeted further with sampling and other finer-scale mapping techniques.





Synchrotron X-ray fluorescence (SXRF) imaging and X-ray absorption spectroscopy (XAS) techniques have the ability to provide information about both the distribution of major and trace elements, including U and Pb, in samples as well as their valence state and speciation. Using XAS methods it is possible to quantify U valence state at concentrations of a few ppm. Synchrotron X-ray beamlines are available through a competitive process of general user proposals (Sutton et al., 2002). These brighter sources offer micron sized resolution for low ppm levels of the elements of interest. Emerging tender energy

spectroscopy (TES) techniques allow mapping of elements such as Mg and S that are important in carbonates (Northrup, 2019). In addition to element maps which show element distribution and the relationships among elements, spots can be chosen for spectroscopy, providing mineralogy (Mg, Ca) and redox (U, Fe, Mn etc).

    Bench-top microscale XRF mapping (μXRF) of major, minor, and trace elements by energy-dispersive spectroscopy permits geochemical characterization of samples at the micron to decimeter scale. X-ray focusing optics, rather than collomation, allow

high spatial resolutions (ca. 15-25 μm) with laboratory X-ray sources. μXRF has been used for chemical imagining in diverse fields, including materials characterization, archaeology, and earth sciences (de Winter et al., 2017; Allwood et al., 2018; Katsuta et al., 2019; Haschke , last; de Winter and Claeys, 2017), and related instruments will be deployed on interplanetary spacecraft (Williford et al., 2018). Qualitative and semi-quantitative imaging and standardless analysis is rapid and versatile, but quantitative composition determination of carbonate rocks is challenging due to the X-ray-transparent carbonate matrix

(de Winter et al., 2017). As shown here, in some samples with high U concentrations, U abundance can be mapped with appropriate analytical conditions, such as incident radiation filtering, multiple beam dwell passes, acquisition under vacuum, and preparation of thick samples that permit excitation of all available U atoms through the sample depth. Data presented here was acquired using an M4 Tornado (Bruker) operating with a 30 W Rh x-ray tube excited at 50 kV and focused to a 17 μm spot (1 $\sigma$ of incident energy at Mo-K$\sigma$) on flat, polished, thick (> 2 mm) samples under 20 mbar vacuum. Fluorescence energy was

detected with two 60 mm$^2$ XFlash silicon drift detectors (Bruker), energy spectra were deconvolved with Bruker software, and total counts in each emission line region were exported for plotting and visualization using MATLAB (Mathworks).

## 2   Case Studies

We use three natural carbonates that have potential as LAcarb standards to illustrate imaging and spectroscopy techniques. We also discuss what is known of the geology and geochemistry to provide context for other samples that might be dated.

**2.1   Late Permian Marine Cement- Permian Reef Complex**

Roberts et al. (2017) offered WC-1 as a primary standard for LAcarb. This late Permian marine cement sample is reasonably homogeneous with high enough U/Pb and radiogenic enough Pb isotopes to make it a suitable reference material. A brief description of the field relationships and U-Pb characterization were presented in Roberts et al. (2017), but here we expand on this to offer additional insight into the standard as well as to highlight characterization techniques that are relevant to any

carbonate study. Similar buildups and cements are described from important oil plays in Kazakhstan (Dickson and Kenter,



2014), and comparisons have been made to Precambrian marine aragonite cements (Grotzinger and Knoll, 1995). Cements of this type offer good potential for giving ages of penecontemporaneous marine diagenesis.

The Permian Reef Complex, spectacularly exposed in the Guadalupe Mountains National Park of West Texas, stands in relief primarily due to the type of cements that make up WC-1. Neptunian dikes are found parallel to the reef track, filling
fractures that resulted from tensional stresses as the reef built out into the Permian Basin (Hunt et al., 2003; Budd et al., 2013; Frost et al., 2013). Early marine cements in the reef and within these Neptunian dikes are primarily botryoidal aragonites and are typically a dark brown color. The dark brown color is from organic matter that is occluded in the carbonate such that when broken or sawn these cements smell like hydrocarbons (anecdotally this strong smell is a good criterion for selecting carbonates for dating). The cements have mostly been altered to calcite, though Chafetz et al. (2008) found original aragonite in similar
cements. Calcite with the greatest alteration is light brown and has less favorable U/Pb (Jones et al., 1995). Petrographically these cements preserve details of the original fibrous aragonite as inclusion trails. WC-1 (Walnut Canyon) comes from a Neptunian dike in the Tansil Member equivalent of the of the Permian Reef Complex. This Neptunian dike is exposed in Walnut Canyon just outside the Guadalupe Mountains National Park entrance. The botryoids that fill the Neptunian dike were made of radiating aragonite needles that bundle into cm scale packages. The botryoids grow atop one another from both sides of the dike (Fig.
2). At the hand specimen scale, cross-cutting white veins are obvious ((Fig 3). Other veins are smaller but can be detected with petrography and element mapping (Fig. 4). The Permian Reef Complex has seen many episodes of diagenetic fluids, including meteoric fluids accompanying the Basin and Range extension that exhumed the reef in the Neogene (Bishop et al., 2014; Loyd et al., 2013; Scholle et al., 1992; Budd et al., 2013).

Cathodoluminescence (CL) has been a go-to test for diagenesis, and when an activator such as Mn is present, this often gives
phenomenal images that illuminate alteration. However, it is also understood that Fe quenches luminescence, so the Mn/Fe ratio matters for understanding what luminescence or non-luminescence means (Barnaby and Rimstidt, 1989). In the case of Walnut Canyon, the well preserved botryoids are non-luminescent and cements that line the botryoids are brightly luminescent (Fig. 4). Even through the botryoids are non-luminescent with excellent textural preservation, the wide range of Sr concentrations across the botryoids shows that they have been variably altered (Fig.4, Fig. 5, Fig. 6). Element imaging is complimentary
to petrography, and while petrographic techniques like CL can provide qualitative information on Mn and Fe, XRF and LA imaging techniques give quantitative information on the Fe/Mn, Sr/Ca and other ratios and element concentrations, providing far more insight into the carbonate diagenesis. μXRF scanning is reasonably accessible, and we suggest that depending on the original mineralogy, elements like Sr (Fig. 4, Fig. 5) could be mapped and registered so that only the best preserved parts of a rock slab could be targeted for a LAcarb standard. This should greatly reduce the scatter that is currently seen in the WC-1
standard.

The Walnut Canyon sample has incredible preservation of the original fibrous aragonite texture even though it is almost entirely converted to calcite. This neomorphic replacement may facilitate retention of U (Kelly et al., 2003), or perhaps the organic matter (that gives the sample its color and pungent smell when broken) complexed U and retained it in the calcite. Remarkably, Sr is lost well before U in this system as evidenced by the rather homogeneous U concentration at about 4.5 ppm
(which makes this a good U/Pb standard) and the highly variable Sr (Fig. 4). Isotope dilution of this sample doesn't give much





of a spread in U/Pb, so to maximize the spread a range of aliquots from dark brown to light brown were utilized (Roberts et al., 2017). Like the result reported in Jones et al. (1995), this produced an age that is nominally younger than the age of the reef based on dating of ash beds (Wu et al., 2020) (254 instead of 257 Ma) , but not outside the uncertainty. The cements postdate reef cementation because they fill fractures, but contain internal sediments and likely cemented penecontemporaneously with

reef deposition. The $^{87}$Sr/$^{86}$Sr ratios range from 0.7069-0.7072 (Fig. 7), with an average of 0.706930(69) for samples that make up the bulk of the dike. These values are similar to what Chafetz et al. (2008) reported from samples with primary aragonite from the Tansil equivalent teepee structures from the Permian Reef Complex, and are consistent with the Capitanian age of the Tansil reef. Samples at the center of the dike (youngest position) give $^{87}$Sr/$^{86}$Sr near 0.7072 (Fig. 7). Without more work it is not possible to know if the variation with position from the host is due to diagenesis or if it captures real changes

in seawater through the growth of the cements. Rasbury et al. (2004) dated similar cements from the algal mounds of the Laborcita Formation in the Sacramento Mountains of New Mexico, and obtained ages and $^{87}$Sr/$^{86}$Sr that was also consistent with the age of deposition.

Laser ablation maps of WC-1 were presented in Roberts et al. (2017). Sr concentrations ranged up to 7000 ppm, and regions with high Sr have high U. Magnesium is elevated in veins along with V and Mn, as is also shown with μXRF(Fig. 5) and TES

(Fig. 8). While LA and μXRF do not offer the high resolution of synchrotron XRF, they do provide details at the scale that LA dating could be accomplished and few elements are inaccessible with quadrupole mass spectrometers.

High resolution, on the fly element mapping in the tender energy range (TES) allows us to examine U, Mg, Sr and S in the Walnut Canyon sample. Comparing U, S, and Sr in a RGB map (Fig. 8) illustrates the spotty retention of Sr and S relative to U. As seen in the μXRF images (Fig. 5), Mg is elevated in veins and is thus introduced by later fluids. U(M5) XANES shows that

the botryoidal cements have entirely oxidized U (Fig. 9). Uranium concentrations in this samples are less than 5 ppm, and the ability to explore U oxidation states at this concentration is a major advance. An important factor to be considered for exploring for favorable U/Pb in carbonates is how U is incorporated. In the Walnut Canyon case we hypothesize the uranyl was brought to the site of precipitation by an oxidizing fluid (seawater) and structurally incorporated in aragonite which does not exclude U (Reeder et al., 2001; Kelly et al., 2003). Through neomorphism to calcite, U was left behind because active functional groups

such as carboxyl have a high affinity for uranyl. Perhaps calcite grew around these complexed U ions preserving near original U concentrations. In contrast, Sr, which is also strongly partitioned into aragonite but has a lower affinity for calcite, was lost to the fluid because it does not have an affinity for organic matter.

## 2.2  Middle Miocene Tufa- Barstow Formation

The Middle Miocene Barstow Formation crops out in the Rainbow Basin near Barstow California. Tufa mounds are found near

the top of the Owl Conglomerate (Cole et al., 2004, 2005) and are assumed to have formed where springs entered a lake similar to Mono and Pyramid Lake tufa towers. These tufa deposits have more than 100 ppm U, several ppm Pb and favorable U/Pb systematics (Cole et al., 2005). Here we build on the work of Cole et al. (2004, 2005) with additional synchrotron imaging, μXRF imaging and spectroscopy techniques. The sample that we are using to illustrate these tools for characterization is large enough to be suitable for distribution as a secondary standard for LA carbonate dating (Fig. 10). Most of the samples used in





the (Cole et al., 2005) study were small slabs from Vicki Pedone (Cal State Northridge) are too small to to distribute beyond
      our lab, so we are characterizing a new sample that is large enough to distribute widely. The tufa samples ranged in age from
      14.8-17 Ma (Cole et al., 2005). Isotope dilution on this sample is under way and will be completed when Covid quarantine is
      lifted. However, based on the published ages the sample is approximately 15 Ma, and has $^{238}$U/$^{206}$Pb between 150-450.

      Becker et al. (2001) showed that layers in the Barstow tufa deposits have a pattern of low to high U concentrations across

laminae revealed by fission track mapping. Uranium is lower in the beginning of a laminae where larger crystals of calcite
      formed, and is higher in micritic calcite which forms the caps of laminae. Becker et al. (2001) reasoned that the laminae reflect
      seasonal increases in fresh water supply which delivers Ca to a body of water with high alkalinity similar to the Great Basin
      Lakes. Phosphor imaging shows that in addition to these fine scale trends observed with fission tracks, there are mm to cm
      scale layers of higher and lower U concentrations (Cole et al., 2004).

Synchrotron and µXRF imaging of Sr and Mn faithfully matches the primary layering, demonstrating that later alteration
      has been minor (Fig. 11, Fig. 12). A comparison of U-L $\alpha$ maps with and without an incident radiation filter shows that
      the filter eliminates artifacts that result from scatter in pores, and returns an accurate map of the U (Fig. 13). The filtered U
      µXRF map shows that U is elevated in the micritic areas of the tufa. µXRF maps of Al and Si show that large pores that are
      typical of the tufa deposits have clay minerals lining them (Fig. 12. Fe, Cu and Zn are also elevated in the pores. A map of Ba

closely matches the S map suggesting that barite may be present. Strontianite is found in economic abundance in the Barstow
      Formation (Knopf, 1917). The extremely high Sr concentrations in the tufa calcite (10's of thousands of ppm), suggests these
      fluids were contributing Sr during deposition. The $^{87}$Sr/$^{86}$Sr isotope ratios are similar throughout the sample, with 3 aliquots
      ranging from 0.719877 to 0.721038 (Table 1). Combining the criteria for selection of aliquots for U/Pb, with high Sr content,
      is likely to greatly narrow this range, making this a good Sr isotope standard as well as a secondary U/Pb carbonate standard.

The U in these tufas is in the reduced state U(IV) as shown by measurements at the L3 edge (Cole et al., 2004), and new data
      from the M5 edge (Fig. 14). With the high U concentration, the Barstow tufa could be a standard for U XANES analyses. The
      Barstow tufa calcite is luminescent throughout (Fig. 15), consistent with the occurrence of reduced U in this sample. While
      it is easy to conceive of a stratified lake with reducing bottom waters, reduced U is insoluble in most solutions, begging the
      question of how it is available to go into the calcite. We hypothesize that U(IV) is complexed with some oxyanion in the lake

water (phosphate, bicarbonate, etc) that keeps it in solution and perhaps is also incorporated into the calcite lattice. Elevated
      actinide concentrations are found in the Great Basin lakes, and it is thought that the carbonate alkalinity is responsible for this
      elevation (Anderson et al., 1982; Simpson et al., 1982). The oxidation state(s) of U in the Mono Lake carbonates is not known,
      but we imagine waters with similar chemistry might be responsible for elevated U in the Barstow tufa. While we do not have
      temperatures of formation, the morphology of the tufa is similar to that at Pyramid Lake where warm springs enter the lake

(Cole et al., 2004). Perhaps thermal fluids are important for U(IV) mobility.

      One of the biggest advances in LAcarb dating is a contribution by Drost et al. (2018) which used laser ablation imaging and
      inspection for concentrations or ratios that reflect something that could be considered to be related (like Sr concentrations).
      Pooling of pixels based on this criteria from across the mapped region and binning based on some additional criteria like
      $^{235}$U/$^{207}$Pb gives isochron plots with range of values that is typically greater than, and certainly more filled in than, would be





obtained by spot analyses. The approach is justified because it would not be possible to select pixels based on criteria other than the ratios being plotted, that would have a meaningful isochron result if the assumption were invalid. This work built on a monocle approach introduced by Petrus et al. (2017) which provided a visual tool to easily compare maps, ratios, and patterns (such as REE patterns) that can be defined by the user. This package of tools is available in Iolite3 and Iolite4 and, beyond U/Pb dating, offers tremendous potential to better understand relationships between elements and how that relates to

deposition and diagenesis. Inspired by the Drost et al. (2018) contribution, we did 45 laser ablation lines on the Barstow tufa (Fig. 16). We used Iolite4 (Paton et al., 2011) for the data reduction. Note the laminae are easily visible in the element maps 16. While the U concentration is not as high in the sparry calcite layers, the U/Th and $^{238}U/^{206}Pb$ ratios are much higher than the micritic calcite 16. Similarly, the $^{208}Pb/^{206}Pb$ and $^{207}Pb/^{206}Pb$ are much lower, showing the radiogenic ingrowth of $^{206}Pb$. Sr and Mg concentrations are elevated in the sparry calcite so µXRF maps of slabs would provide a framework for selection of

spots for LAcarb (Fig. 12). We extracted pixels based on the criteria that they were greater than 12,500 ppm Sr. We used this value based on the monocle feature in Iolite4 which allowed us to see that this would eliminate most of the pixels outside the sparry calcite (Fig. 17). Nearly 7000 pixels meet the criteria for being greater than 12,500 ppm Sr and less than 1 ppm $^{208}Pb$. These were subdivided into 90 bins of 70-80 pixels each based on probability of the $^{207}Pb/^{235}U$ ratio e.g. (Drost et al., 2018). When all of the bins are plotted in Iosplot (Ludwig, 2003), there are tails on the low and high ends of the line. Removing 10%

on each end produces a much better line (Fig. 17). The age of 12 Ma is about 20% too young. We did not make a correction because we do not know the age of this sample, but a correction could be made based on the secondary standard and applied to unknowns. The purpose here is not to provide the age, but to show that the sample is very well behaved. Pre-screening with µXRF to establish the best regions to use for LAcarb should produce a reproducible secondary standard for QA/QC.

## 2.3 Middle Miocene Dolomite

A Miocene conglomerate horizon in the Napenagila fossil locality of the Turkana Basin in Kenya is extensively cemented by calcite and dolomite. The dolomite is unique in a number of ways. It forms an isopachous layer on surfaces of fossil trees, showing that it formed in the saturated zone. It has a bladed habit, and is a bright yellow color, becoming darker yellow in the growth direction (Fig. 18). While only an outer crust of the sample has typical bright orange luminescence, the sample glows in the luminescope (Fig. 18). The dolomite has variable trace element concentrations along the growth direction, and

extreme U enrichment in the final phases of growth as shown by synchrotron XRF element mapping (Fig. 19), µXRF (Fig. 20) and laser ablation (Fig. 21). The bladed morphology of the dolomite crystals is strongly emphasized by trace element patterns in the youngest part of the dolomite, and is particularly striking in the very high resolution maps from TES (Fig. 19). U M5 XANES shows that the U in the dolomite is in the reduced state (Fig. 23). The stratigraphic context of this dolomite and other carbonates from the Turkana Basin is being studied by Kevin Hatton and will be presented in a more comprehensive way

elsewhere. Meanwhile, the very well-behaved U-Pb systematics make this a possible dolomite standard, and the XRF mapping and X-ray spectroscopy make it a good example for carbonate characterization. The $^{238}U/^{206}Pb$ ratios for the dolomite sample range from 250-550. Using a selection criteria of Sr concentrations greater than 2500 ppm, and $^{208}Pb$ less than 0.3 ppm produces over 2200 pixels. Binning the pixels into 70 bins produces an age of 15 Ma (Fig. 22). Removing the 'tails' produces





data that plots near the concordia curve but is poorly anchored on the common Pb side (Fig. 22), and we chose not to assume an initial isotope value. The average of the $^{206}$Pb/$^{238}$U ages produced by selection of pixels within polygons on the basis of high U along a crystal is 14.52 Ma (Fig. 22). The average of the $^{206}$Pb/$^{238}$U ages based on binning pixels where Sr is greater than 2500 and $^{208}$Pb is less than 0.3 ppm (which was based on the monocle inspector in Iolite4) is 14.77 Ma (Fig. 22). We are working on isotope dilution of this sample which will provide an age for comparison, but whether the age is correct or not, the U/Pb systematics are very favorable.

$^{87}$Sr/$^{86}$Sr of the aliquots of the Turkana dolomite from the outside (oldest) to the inside (youngest) are indistinguishable from each other at 0.703306 (Table 2). With the well behaved U/Pb systematics, high concentrations of reduced U and homogeneous $^{87}$Sr/$^{86}$Sr, this sample could be a standard U/Pb dating, Sr isotopes, and synchrotron U spectroscopy.

## 3 Summary

Field relationships and petrography allow us to consider the relative timing of carbonate precipitation and alteration, the tools for demonstrating these relationships are an important first step to any study. XRF mapping (both µXRF and synchrotron XRF) allows us to establish domains within rocks that are equivalent and to work out details of the fluids that were responsible for carbonate precipitation. Details such as the U oxidation state, and relationships to other elements helps to further constrain the fluids that have produced the final product. While elevated U is not necessary for U/Pb dating, samples that are amenable to LA Carb dating typically must have ppm levels of U. The tools we have described here allow us to provide a spatial context that with petrography and field relationships are foundational for putting U/Pb ages into the overall geologic context.

*Author contributions.* ETR led the effort to review the literature on carbonate characterization. TP did the µXRF imaging of the carbonate standard materials. PN RT, and AL led the efforts for the synchrotron XRF and XANES measurements. KW and KH led the laser ablation mapping efforts. KW did the Sr isotope analyses. JMC did the work on the Barstow tufa. All of the authors contributed to writing the manuscript.

*Competing interests.* We declare that there are no competing interests.

*Acknowledgements.* Portions of this work were performed at beamline X26A of the National Synchrotron Light Source (NSLS), Brookhaven National Laboratory. X26A was supported by the Department of Energy (DOE) - Geosciences (DE-FG02-92ER14244 to The University of Chicago - CARS). Use of the NSLS was supported by DOE under Contract No. DE-AC02-98CH10886. Portions of this research were also performed at the Tender Energy Spectroscopy (TES), X-ray Fluorescence Microprobe (XFM), and Submicron Resolution X-ray Spectroscopy (SRX) beamlines and used resources of the National Synchrotron Light Source II, a U.S. Department of Energy (DOE) Office of Science User Facility operated for the DOE Office of Science by Brookhaven National Laboratory under Contract No. DE-SC0012704.



Construction of, and work at the TES Beamline was partly funded by the National Science Foundation, Earth Sciences (EAR-1128957), NASA (NNX13AD12G) and the Department of Energy, Geosciences (DE-FG02-12ER16342). µXRF data was acquired in the Caltech GPS Division Analytical Facility with the support of Chi Ma, John Grotzinger, and the Simons Foundation. We thank Stephen Cox, Frank Sousa,

Elena Steponaitis and Sidney Hemming for the discovery of the dolomite in the Turkana Basin and acknowledge funding from the Columbia University Global Center for field work. Numerous discussions with Bruce Ward have greatly improved our understanding of carbonates.



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



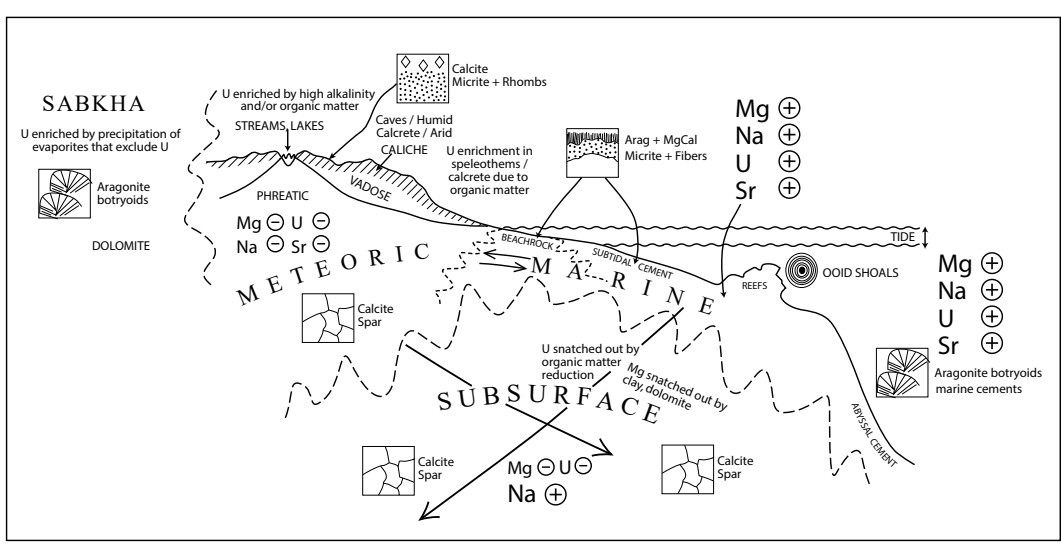

**Figure 1.** Cartoon of the variety of carbonate producing environments with arrows indicating the direction the elements would change through diagenetic alteration (Rasbury and Cole, 2009) We would like to see U concentrations and oxidation states be an important component in understanding fluids and diagenesis

.



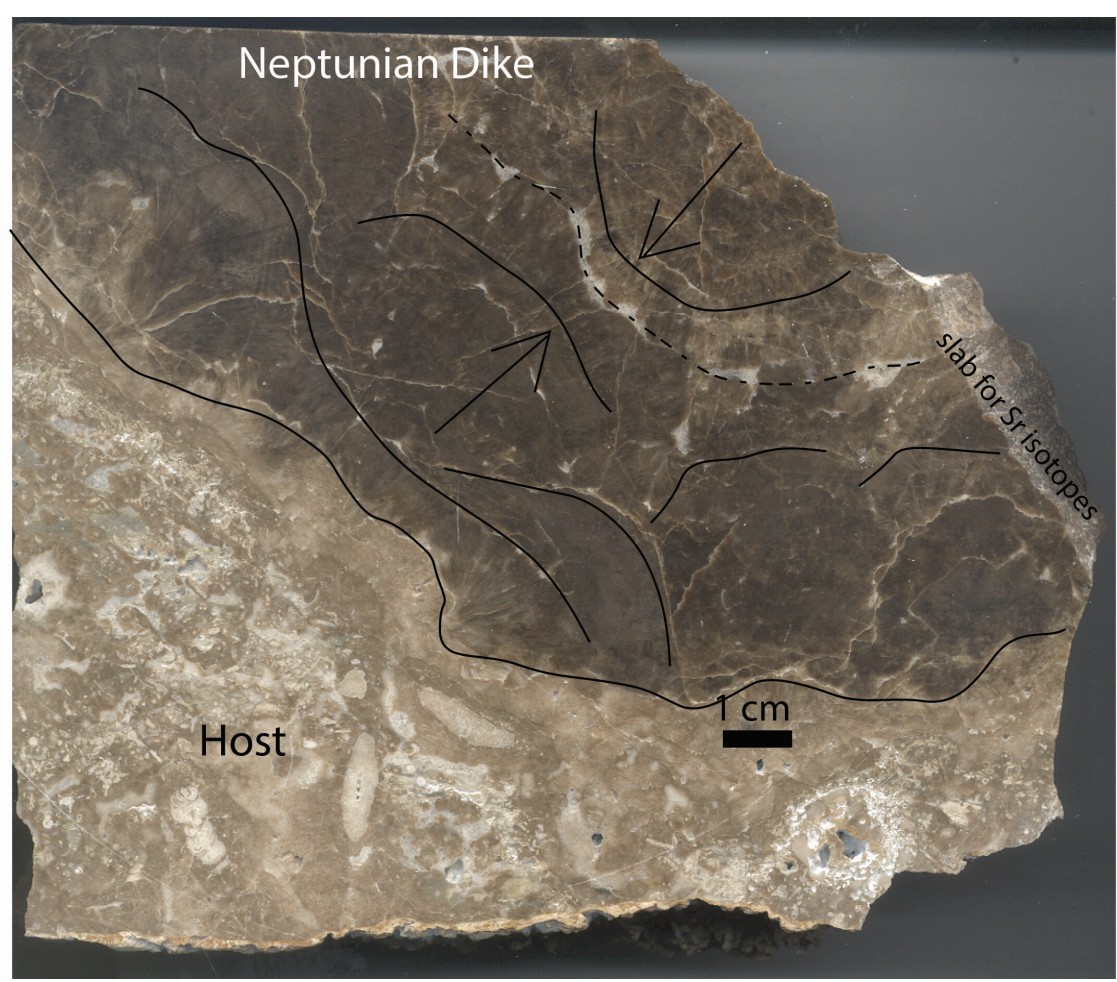

**Figure 2.** Large hand sample from a Neptunian dike at Walnut Canyon. The lighter brown limestone on the left is the reef rock with abundant large forams visible here. The dark brown botryoids can be seen growing from both sides of this oblique slab through the dike carbonate. Curved lines show approximate botryoid boundaries and arrows point in the growth direction. The dashed line is approximately the center of the dike. A white calcite cement lines many of the botryoids and occurs in the center of the dike. A slab from the host to the center was analyzed for the Sr isotopes presented in this paper.



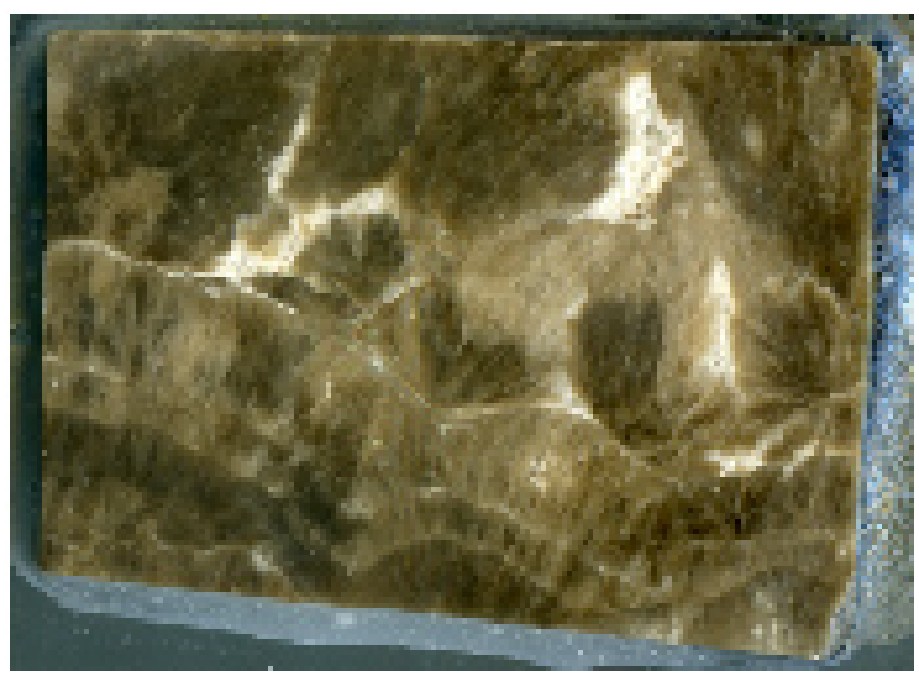

**Figure 3.** Polished slab from the Walnut Canyon Neptunian Dike sample used in this study to illustrate various tools for characterization. This is approximately 3 cm across. This is a similar size to that being distributed a a primary standard (Roberts et al., 2017). Note the variability of color from very dark to light brown, and white veins. The fibrous botyroidal texture is visible even at this scale, and complexities due to veins should be easily avoided when using this as a LAcarb standard.

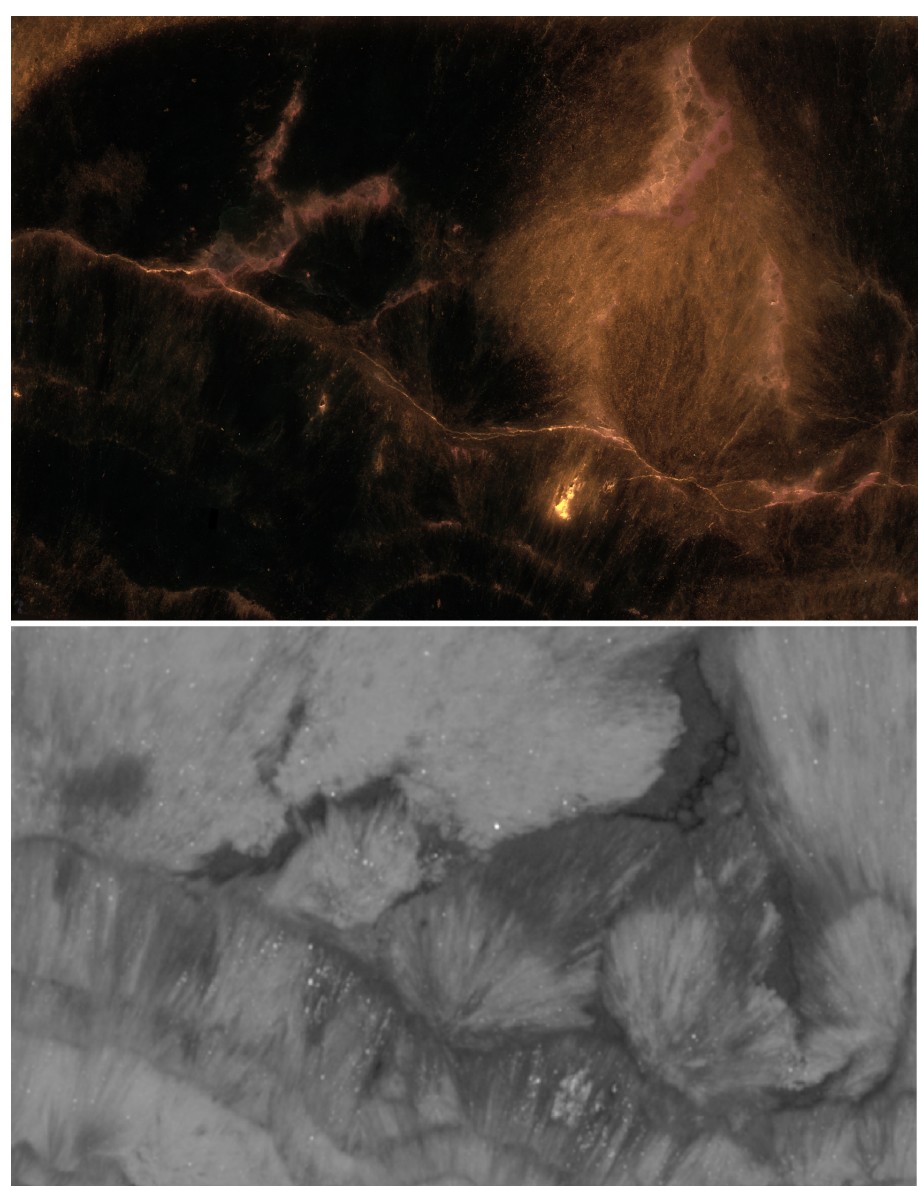

**Figure 4.** Upper image is a photomosaic of 639 CL photos. The bottom imaage is a microXRF map of Sr (details in the next figure caption). Areas that are brightly luminescent have low Sr. Boundaries between botryoid layers are luminescent (and white so easy to avoid) and should be avoided when using this sample as a standard. This microXRF map of Sr beautifully shos the texture of the sample. We suggest that this easy screening technique could be done on all slabs being used for a LA standard, and that this could greatly reduce the scatter that researchers are finding for this standard.





**Figure 5.** Relative intensities of emission regions for each element detected in Walnut Canyon, imaged at 20 µm pixels with 1.5 s.d. gaussian blur. Dwell time was 30 ms/pixel for maps with no incident radiation filter (Mg, Al, Si, S, Ca). Dwell time was 60 ms/pixel for maps with incident radiation filtered with 100 µm Al, 50 µm Ti, 20 µm Cu foil (Cl, Ti, Mn, Fe, Sr, U). Each panel is gamma and contrast adjusted to emphasize gradients between textures.

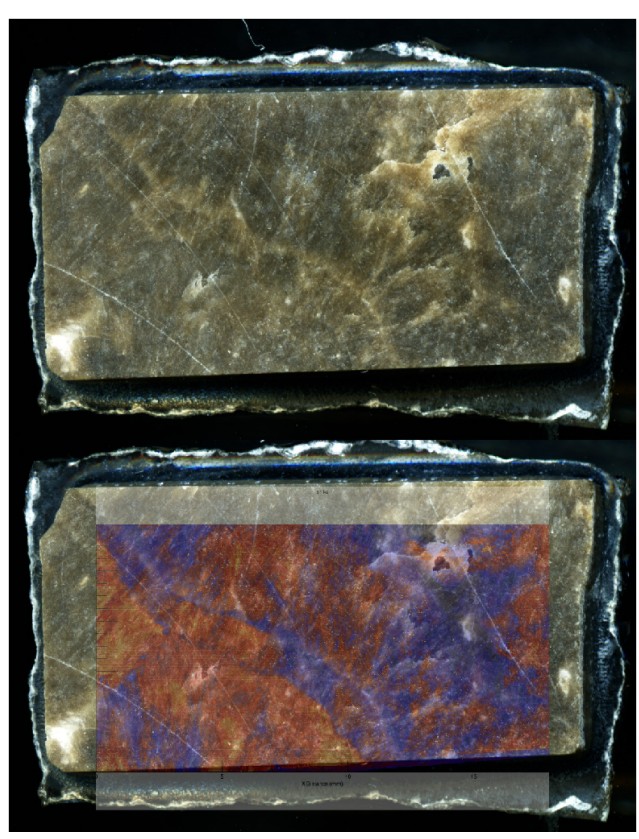

**Figure 6.** Synchrotron XRF Sr maps from beamline X26A NSLS Brookhaven National Lab overlain on the slab that was imaged. This XRF map was accomplished by on the fly scanning. To obtain U maps, we would have had to scan the same area above and below the binding energy for U and make a difference map because the SrKa peak is so large that the shoulder overlaps with the far less abundant UL3 peak. Still, based on the wealth of background information, we know that Sr is lost before U, so that focusing on botryoids with elevated Sr should produce the best areas for LAcarb standardization. This type of image can be registered for laser ablation so that only the most favorable areas are used for standardization.



**Figure 7.** This graph shows Sr isotopes from the host to the center of the Neptunian dike. The pink symbols represent two independent analyses from slabs that are quasi-equal distance from the host, and the blue is the average of those analyses. While showing variability, the dark brown calcite has $^{87}$Sr/$^{86}$Sr like Capitanian seawater with an average of 1-5 of 0.706930(69). The samples from positions 6 and 7 are a light brown color and a boundary is clearly visible in 6. These may represent later fluid alteration, or an extended marine cement history. This is presented because we suggest that this sample may also be a useful Sr isotope standard. The slabs are about 0.5 cm across.

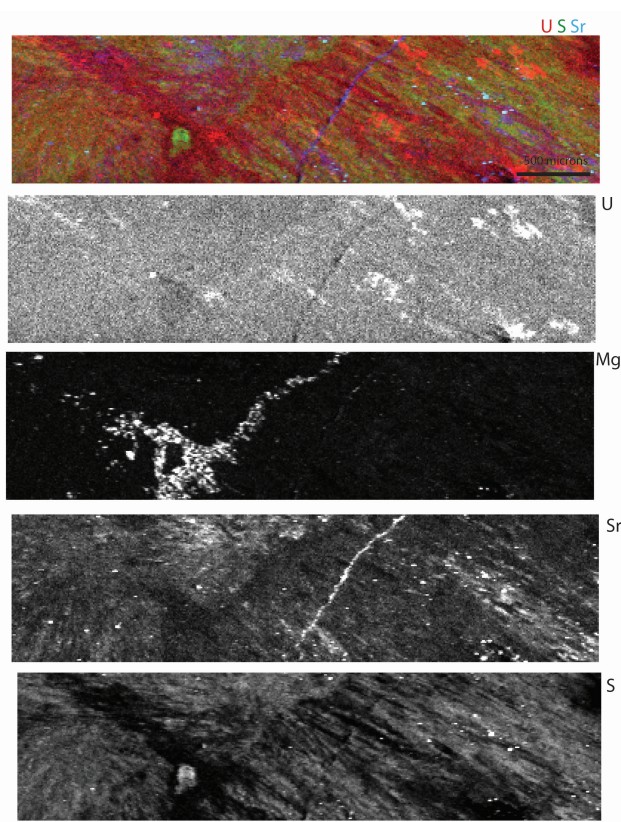

**Figure 8.** Tender Energy Range (TES) synchrotron XRF maps. The panels are labeled with the elements they represent. The top map is a RGB map where red is U, green is S and blue is Sr (note scale bar on the bottom right of the RGB image is 100 microns). The brightest areas in the U map are artifacts from diffraction. The best preserved areas are an olive green.



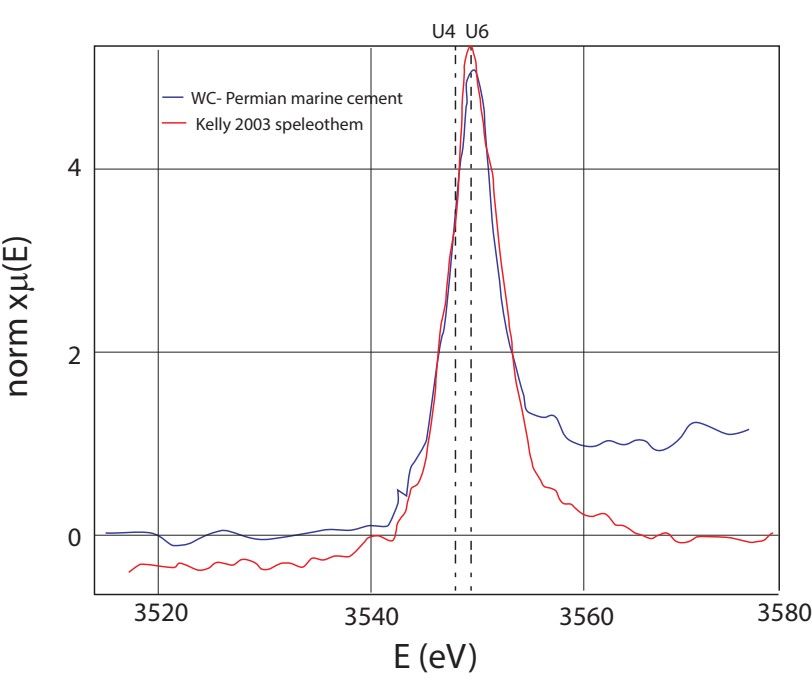

**Figure 9.** M5 edge X-ray absorption spectroscopy, 20 scans ( 15 sec each) showing that uranium is in the oxidized state. The TES beamline has a monochrometer that is very stable, allowing hours worth of scanning without drift. Speleothem from a cave in Austria Kelly et al. (2003) that has U(VI) for reference.



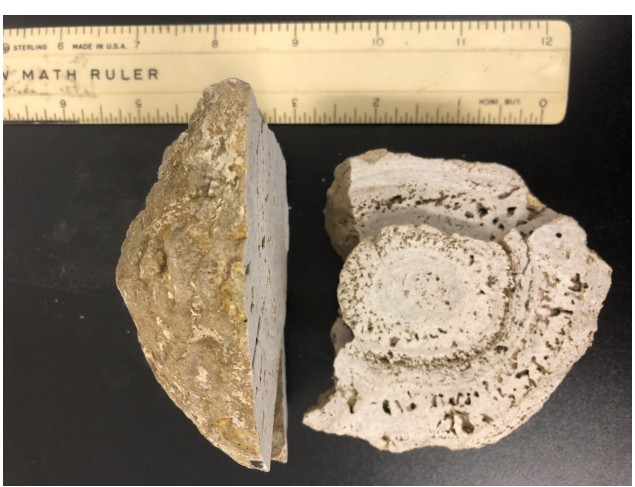

**Figure 10.** Hand specimen of a Barstow Formation tufa sample that is large enough to slab and share as a secondary LAcarb standard.

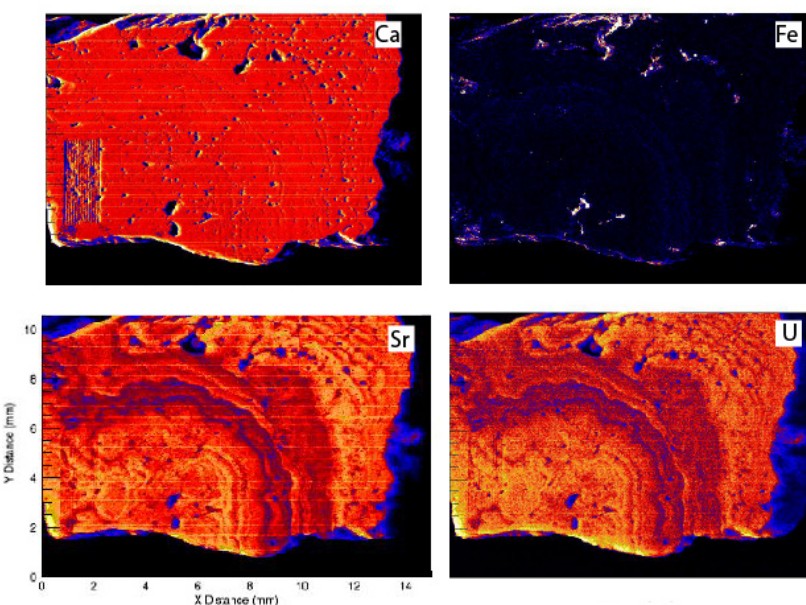

**Figure 11.** On the fly synchrotron XRF maps of Ca, Fe, Sr and U for a slab of Barstow tufa. This in not the sample that is being characterized for a LAcarb standard, but it serves to show the relationships between U and Sr.



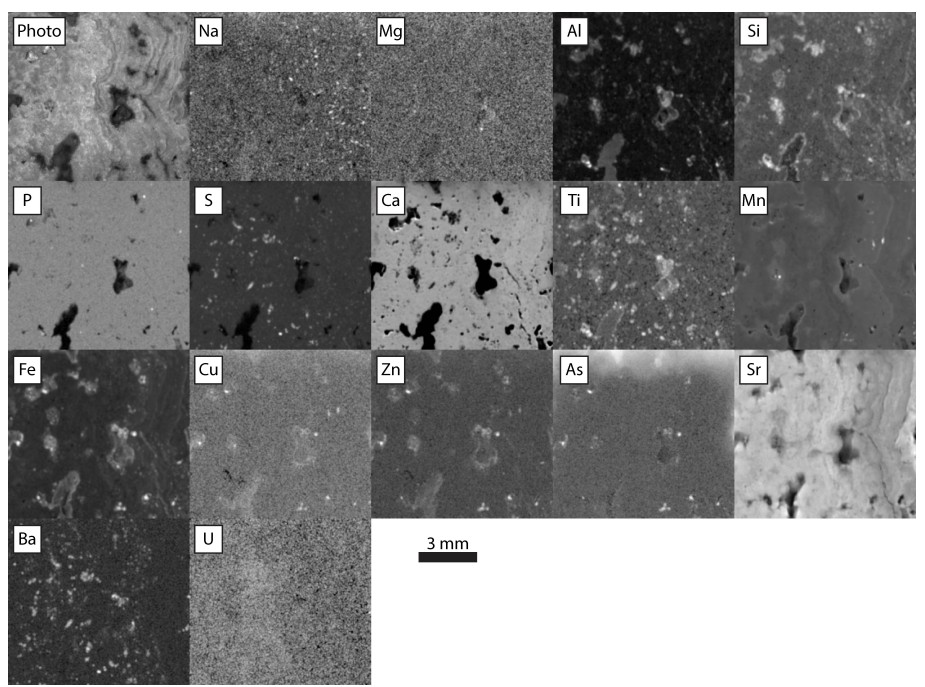

**Figure 12.** Relative intensities of emission regions for each element detected in Barstow tufa, imaged at 25 μm pixels with 400 ms/pixel dwell time and 1.5 s.d. gaussian blur. Each panel is gamma and contrast adjusted to emphasize gradients between textures. Na through Fe data collected with no incident radiation filter. Cu through U data collected with incident radiation filtered with 100 μm Al, 50 μm Ti, 20 μm Cu foil.

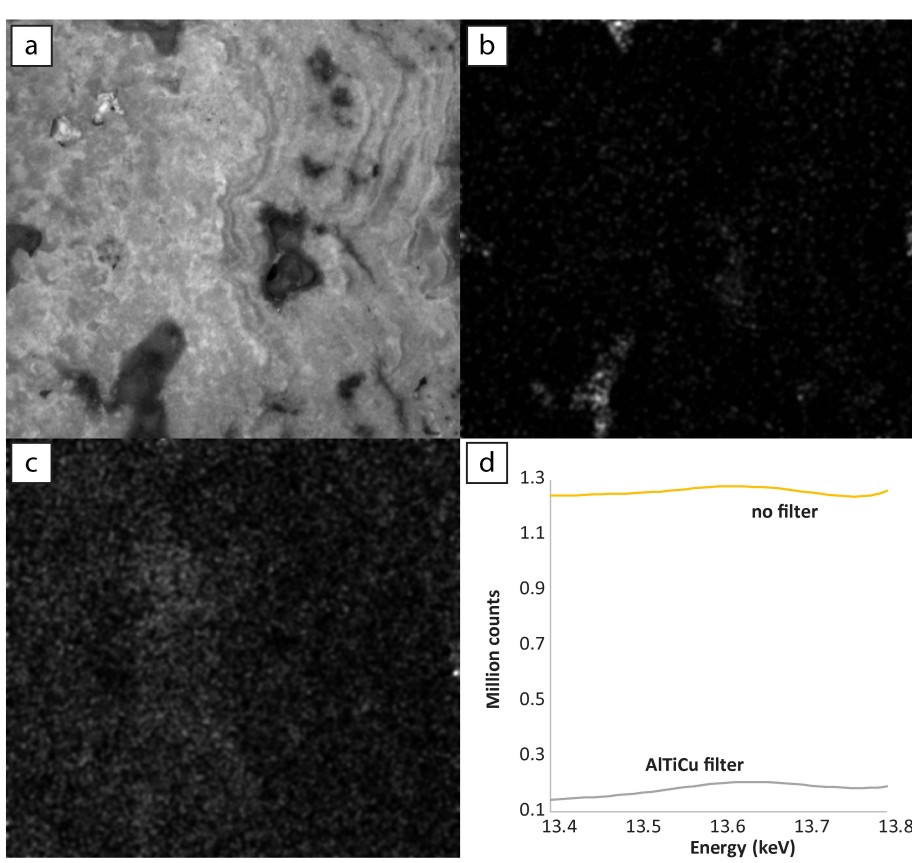

**Figure 13.** Comparison of U-L $\alpha$ maps of Barstow tufa, imaged at 25 μm pixels with 400 ms/pixel dwell time and 1.5 s.d. gaussian blur. (a) Sample image. (b) No incident radiation filter. (c) Incident radiation filtered with 100 μm Al, 50 μm Ti, 20 μm Cu foil. (d) Sum spectra of total counts in U-L $\alpha$ line region.

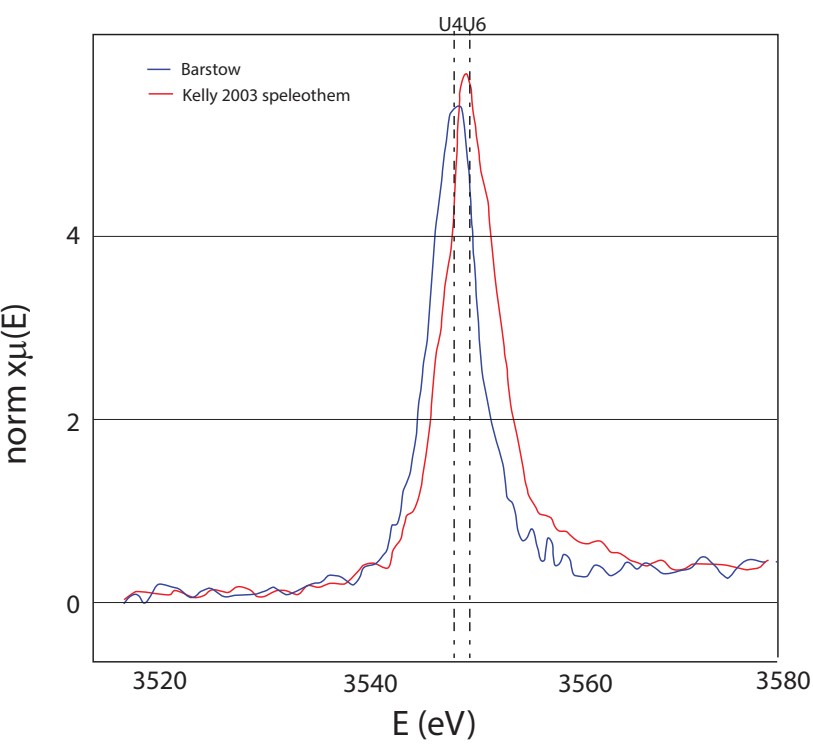

**Figure 14.** Uranium M5 edge spectroscopy showing that uranium is in the reduced state; for reference, the U(VI) speleothem of (Kelly et al., 2003). Both measured at the TES beamline as described in Figure 9.



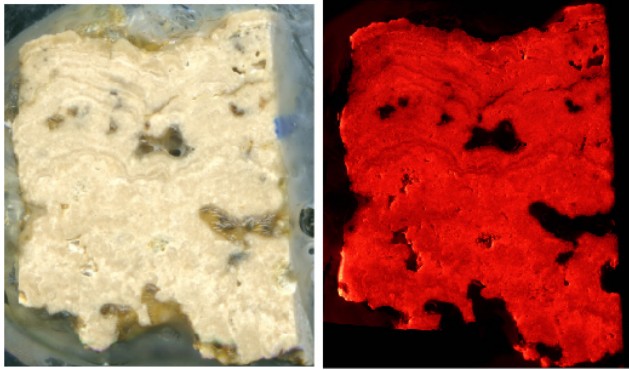

**Figure 15.** Slab from the Bartow Formation tufa sample that is being characterized for a possible secondary LAcarb standard. The fine laminae are like those described by Becker et al. (2001). The left image is a scan of the slab. The right image is a CL photomosaic from 360 images. Generally, the more micritic areas are brighter luminescence and the sparry layers have a duller luminescence.





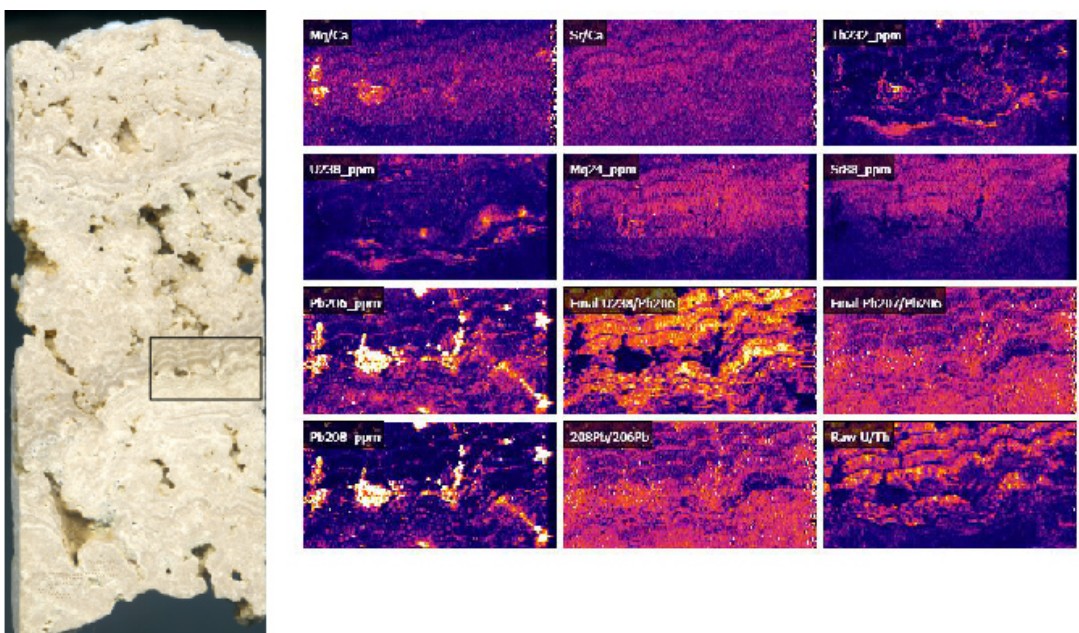

**Figure 16.** Slab of the Barstow tufa, approximately 1 cm wide. The box shows the location of the LA maps.

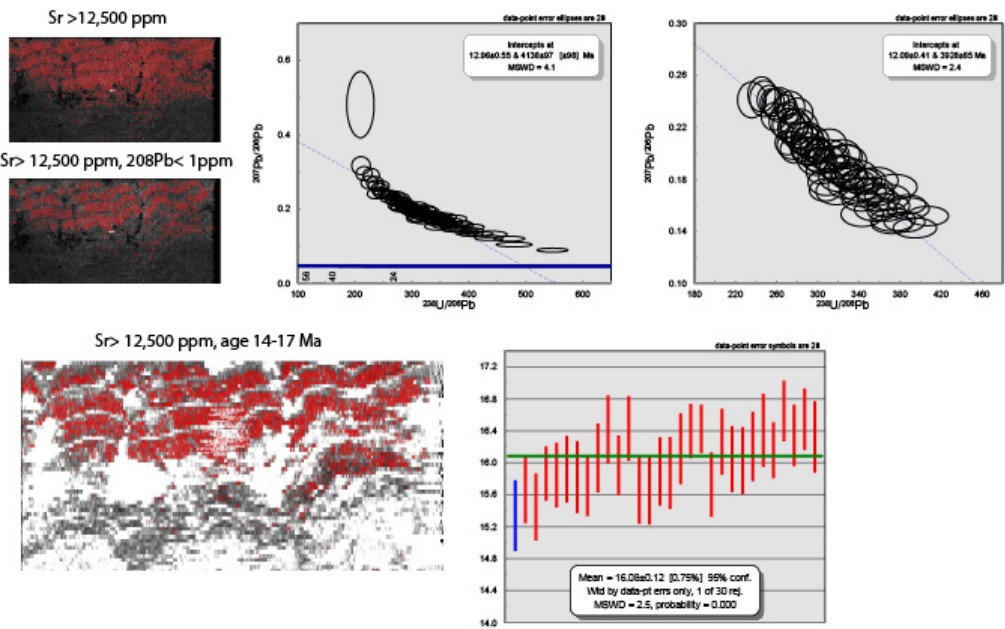

**Figure 17.** Sr concentration map used to select pixels for the sparry calcite in the Barstow tufa. In the top left map, the red marks the pixels that are over 12,500 ppm Sr. The map below, has the same critera plus the $^{208}$Pb is less than 1 ppm. The first isochron plot is a binning of all of the pixels that meet this second critera without removing tails. The second isochron has the top and bottom 10% tails removed. The bottom map is the $^{206}$Pb/$^{238}$U age map and the red pixels are greater than 12,500 ppm Sr and between 13-17 Ma. The probability plot is based on binning those pixels.



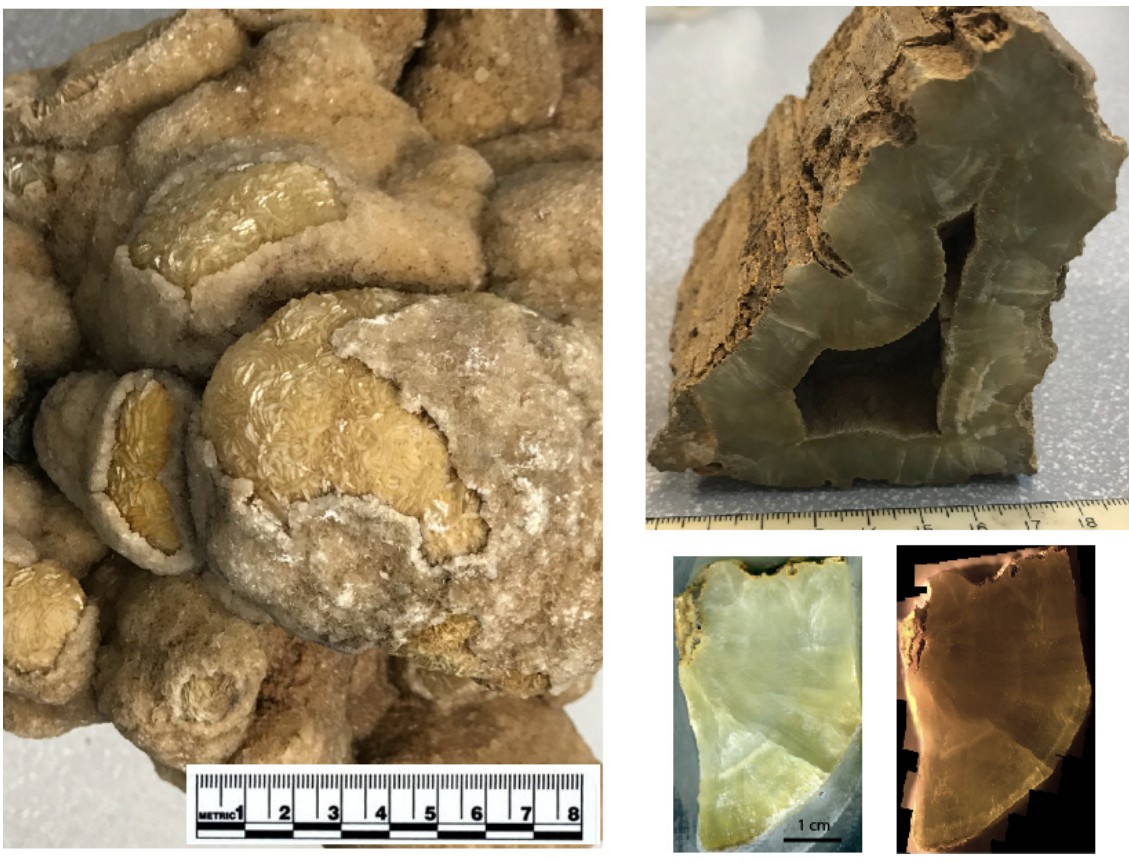

**Figure 18.** Hand samples of the Turkana dolomite sample, both came from an area where carbonates were coating tree trunks. The big sample is from a very large cavity that was filled by calcite, then the yellow dolomite that is peeking out and finally by a coating of grey calcite. The smaller triangular shaped sample has a bark texture on the outside and appears to have precipitated in the cavity between three logs. The thickness of the dolomite layer is about 2 cm and it coats every surface in this paleo log pile. The small slabs come from the smaller of the two samples. The image on the left is the scanned slab used for XRF studies, and the one on the right is a CL photo mossaic with 459 photos.





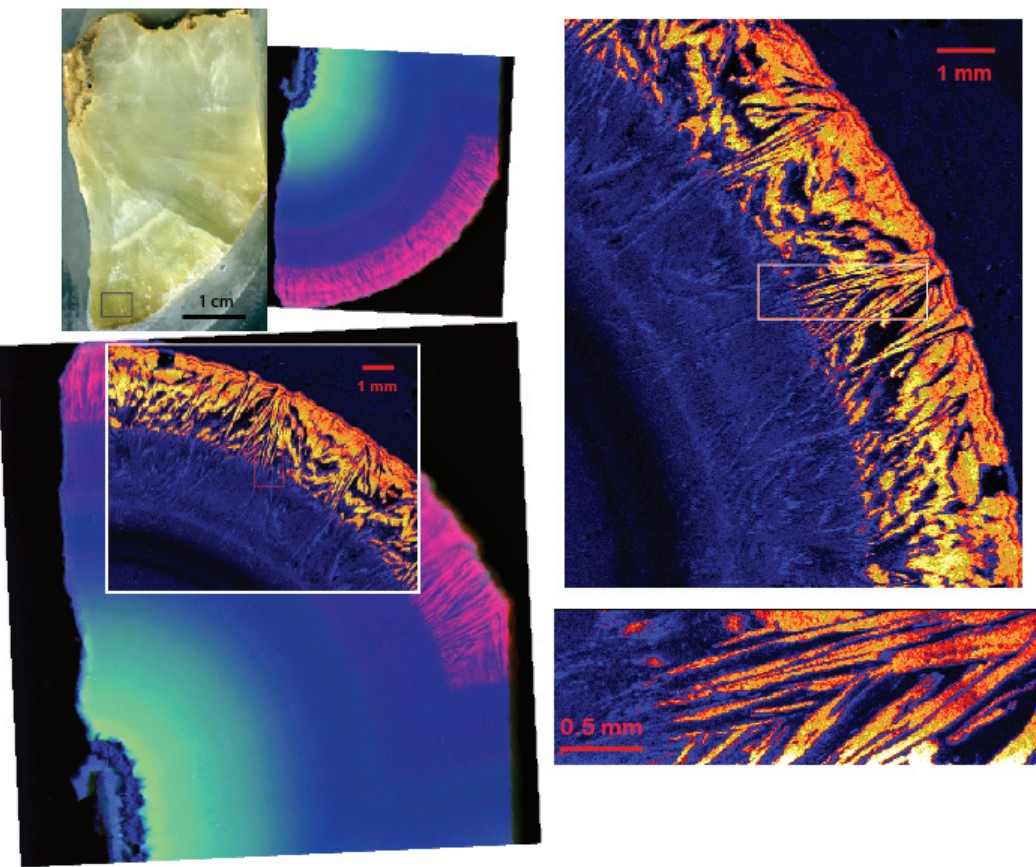

**Figure 19.** Slab of the dolomite with on the fly synchrotron XRF images. The rectangle at the bottom of the sample is the approximate location of the LA maps in Figure 21. Directly to the right of the slab image is a RGB (USrCa) map from beamline XFM at the NSLS II. The images on the right are maps of U from TES. On the lower left, the TES U map is overlain on the same area that was mapped at XFM. The XFM mapping was done at 17 keV with a 10 micron beam and a 10 micron step size, and a counting time of 0.1 seconds. The large TES map was made at 3550 eV with an 18 micron beam, 20 micron pixels, counted for 0.3 seconds. The detail map on the bottom right was also made at 3550 eV with a 4 micron beam, 3.75 micron pixels, counting for 0.3 seconds.





**Figure 20.** Relative intensities of emission regions for each element detected in Turkana Dolomite. Each panel is gamma and contrast adjusted to emphasize gradients between textures. (a) No incident radiation filter, imaged at 25 μm pixels with 120 ms/pixel dwell time and 1.5 s.d. gaussian blur, with box indicating region re-mapped with filter shown in panel b. (b) Incident radiation filtered using 100 μm Al, 50 μm Ti, 20 μm Cu foil, imaged at 20 μm pixels with 900 ms/pixel dwell time and 1.5 s.d. gaussian blur.



**Figure 21.** Laser Ablation ICPMS maps of the TBW dolomite. U is high concentration, while Pb and Th are low. The REE are also low abundance. The U/Pb ratio is very favorable.





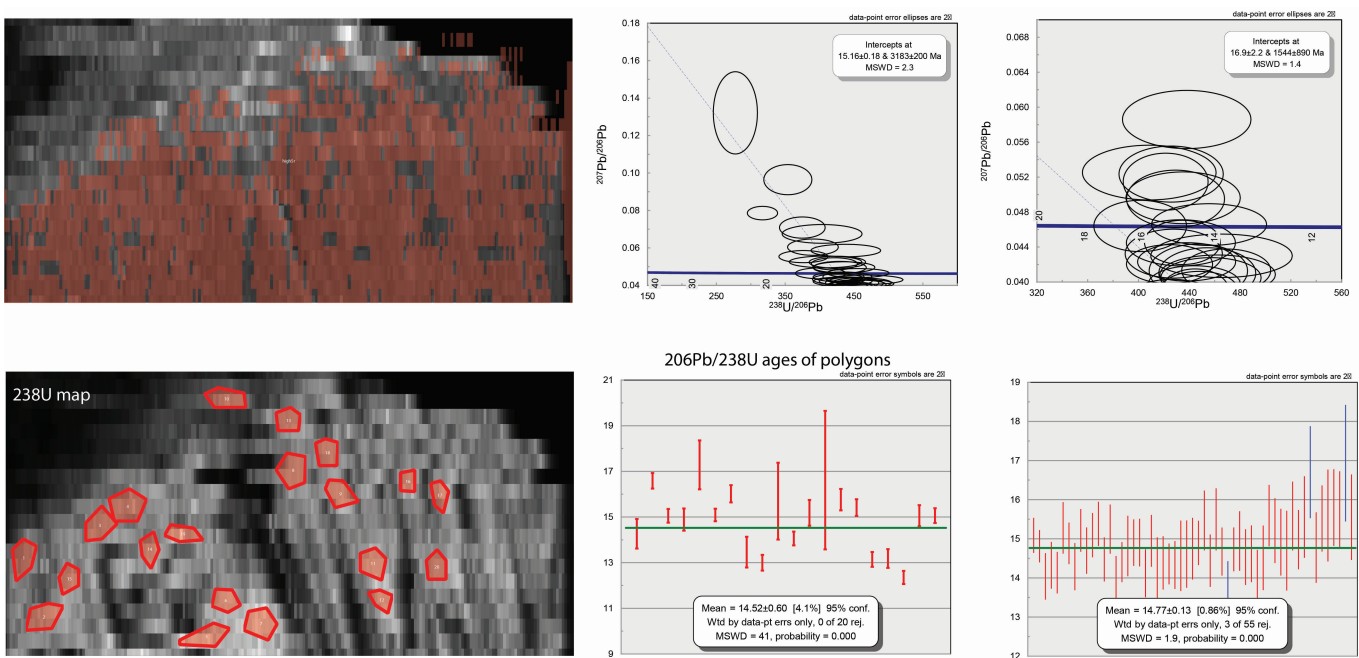

**Figure 22.** Map of U. Regions selected based on U concentration. 206Pb/238U ages of each region shown on a probability plot.







**Table 1.** $^{87}$Sr/$^{86}$Sr from aliquots of the Barstow tufa run by Thermal Ionization Mass Spectrometry

| sample | $^{87}$Sr/$^{86}$Sr | 2SD |
|--------|---------|----------|
| BT1 | 0.720423 | 0.000011 |
| BT2 | 0.721038 | 0.000011 |
| BT3 | 0.719877 | 0.000011 |





**Table 2.** $^{87}$Sr/$^{86}$Sr from aliquots of the Turkana dolomite run by Thermal Ionization Mass Spectrometry

| sample | $^{87}$Sr/$^{86}$Sr | 2SD |
|--------|---------------------|-----|
| TBWi-1 | 0.703309 | 0.000011 |
| TBWi-2 | 0.703299 | 0.000011 |
| TBWo-1 | 0.703306 | 0.000011 |
| TBWo-2 | 0.703312 | 0.000011 |