# Peer review of "Tools for uranium characterization in carbonate samples: Case studies of natural U-Pb geochronology reference materials"

_Geochronology, 2020_

## Referee Comment (RC1) · Anonymous Referee #1 · 10 Aug 2020

The manuscript of Rasbury et al describes a toolkit for U-Pb geochronology and specifically presents characterization of a couple of used or potentially applicable reference materials. The methods are up to date, although some commonly used characterization methods for natural samples are missing. The discussion about U speciation is interesting but requires some more connection to published findings of high U incorporation of U into calcite. I think this manuscript can be published after some minor revisions. Specifically important is the measurements of U(IV) in the calcites, but some more discussion on how the U speciation affect geochronology considerations and results could be suitable. A minor setback is that discussions of processes of U speciation and incorporation and suitability of and heterogeneity of different reference materials are mixed and make the paper a bit un-distinct in its focus.

[Figure]

Specific comments: Title: It would be good if the title specifies that it is calcite and dolomite that is in focus of this manuscript as there are many other minerals under the "carbonate" umbrella. Title and throughout the MS: consider changing "standards" to "reference materials".

Abstract: It would be helpful if it was stated whether the "split stream" analysis of 87Sr/86Sr together with U/Pb, is done with LA-ICP in the same spot, in nearby spots, or if the 87Sr/86Sr is done on dissolved samples and U/Pb with spot analysis. And, if it is the latter, the later discussion should discuss how the difference in scale of larger dissolved aliquotes match spots, when taking heterogeneity of the material into account.

Line 20-21: "Mixing of fluids can be particularly corrosive or can be responsible for mineral precipitation." please give more details.

Lines 21 and onwards: "Introduction of fluids with different chemistries through uplift or burial can destroy the original carbonate prior to precipitation of a new carbonate or can alter the carbonates visibly or at a microscopic scale." difficult to read, please re-structure, e.g. split into two sentences.

Line 31-32: "Special circumstances are required to have elevated U in calcite because the Kd is less than 0.05." Note that this regards oxic conditions and incorporation of UO2, as established in laboratory (please mention this). There are much higher uptake reported for anoxic natural environments that can be mentioned, which also should be mentioned, although U speciation has not been fully characterized yet.

Line 40-41: "A missing component here is that no lab experiments have studied U(IV) incorporation in carbonates. There are several published examples of natural carbonates with reduced U (Sturchio, 1998; Cole et al., 2004) and we present one more in this contribution." The presentation of U(IV) in calcite is highly relevant and an important contribution. Also here (or only here), it would be good to connect these findings to recent findings of very high U uptake in calcite precipitated under anoxic conditions.

[Figure]

Lines 55-58: "Laser ablation mapping (Woodhead et al., 2010, 2007; Piccione et al., 2019; Drost et al., 2018; Roberts et al., 2020), synchrotron XRF mapping (Cole et al., 2004; Piccione et al., 2019; Frisia et al., 2008; Vanghi et al., 2019), PIXE (Ortega et al., 2005, 2003) and $\mu$XRF (de Winter et al., 2017) are all ways to take the physical observations of different phases to a new level with in situ chemistry, that maps U in particular, along with other elements that provide details of the fluid chemistry." It would also be good to add the FE-EMPA for spatial characterization of presence of U nano-particles in calcite (see Suzuki et al., 2016 SciRep).

Lines 61-65: "Given that the major element composition of seawater has changed appreciably through time (Hardie, 1996; Horita et al., 2002), that meteoric fluid compositions are controlled by water rock interaction (Chung and Swart, 1990) and that deep brines have evolved since deposition in basins (Musgrove and Banner, 1993), we can imagine that there is no such thing as a typical fluid. A holistic approach to dating carbonates should involve an effort to see back to the fluid or fluids that have been responsible for its formation and how they might have changed through the diagenetic history."

At some point in the introduction or in the other text, that surch for growth zonations using SEM, CL, and SIMS C- and O-isotope analyses prior to dating. see Milodowski et al APGEO https://www.sciencedirect.com/science/article/pii/S0883292718301938, and Drake et al NatComm 2019 https://www.nature.com/articles/s41467-019-12728-y. What also could be included in a toolbox for complex samples are fluid inclusion studies that can tell about mixing at particular events and about different fluid flow events by giving T and salinity estimates. Another tool is in situ Sr isotope values of different growth zones of calcite crystals. see Drake et al., 2019 for the latter.

Line 129-130: "The cements have mostly been altered to calcite, though Chafetz et al. (2008) found original aragonite in similar cements. Calcite with the greatest alteration is light brown..." This is a bit unclear, is the light brown calcite the most altered of the calcites, or is it replacing aragonite to a larger degree than other calcite types does?

Line 131: "WC-1", mention "Walnut Canyon" at first mention of this material.

Line 135: Fig 3: quality of figure is not publishable. Maybe this has to do with compression for pdf conversion. The veins are difficult to see.

Lines 139-140: "Cathodoluminescence (CL) has been a go-to test for diagenesis, and when an activator such as Mn is present, this often gives phenomenal images that illuminate alteration." and is also very good for veins and single crystals in open fractures, to track different events of precipitation and/or reacitvation and fluid flow fluctuations.

Line 144: Fig. 4, 5, 6: These are nice figures, some suggestions; 1. however, what is the difference between Fig 4a, and 5a Since fig 4b and "5-Sr" are the same, maybe 4 can be deleted and 4a inserted (if it is not the same as 5-"photo" but in B&W, the caption does not say.) 2. What is roughly the range of Sr concentrations in these figures (4-5)? It would be very interesting to know as there is also clear drop in Ca in the Sr-rich parts, so I assume Sr concentrations are quite high.

Line 160: Fig 7: This figure is nice, but would to readers (such as me) who reads from left to write, be more logic if starting at left (old) towards right (young). How were the aliquots for Sr prepared, micro-drill, or are they in situ analyses using LA-MC-ICP-MS? details can be given here.

Line 160: "...with an average of 0.706930(69)", what is "(69)"? a reference with wrong formatting?

Line 165: "...in seawater through the growth of the cements." Here, fluid inclusions may be useful to give more information on the history of fluids and alteration.

Line 169: TES: define, both in terms of settings and acronym

Line 173: fig 8, add to caption which sample it is

Line 179-180: "Through neomorphism to calcite, U was left behind because active functional groups such as carboxyl have a high affinity for uranyl." So was uranyl complexed to carboxyl in the primary fluid and incorporated into aragonite as carboxyl-complexes. Or did this complexation occur after recrystallization? if the former, could the aragonites have had exceptional U/Sr already at the aragonite stage?

Line 189-191; this sentence is not comprehensive, something is missing. and full reference should not be within ().

Lines 192-193: "Isotope dilution on this sample is under way and will be completed when Covid quarantine is lifted." Such a phrase makes me question if this something that will be included in this MS at revision, otherwise this level of details is probably not adding anything.

Lines 198-199: "there are mm to cm scale layers of higher and lower U concentrations (Cole et al., 2004)." what are these concentrations: "higher" and "lower"?

Line 207: "The 87Sr/86Sr isotope ratios are similar throughout the sample..." this is a bit unspecific. maybe say "show relatively small variability" because there is indeed some variability

Line 209; "...making it a good Sr isotope standard..." I am not certain that the three analyses presented here are enough to state it is a suitable standard for Sr isotope analysis. It is stated that zones with high Sr would narrow the range (also for Sr isotope variability?), I cannot see how this is justified with the data presented, especially since there is no information on Sr-concentrations for table 1.

Lines 214-215: "We hypothesize that U(IV) is complexed with some oxyanion in the lake water (phosphate, bicarbonate, etc) that keeps it in solution and perhaps is also incorporated into the calcite lattice." here it would be good to show some thermodynamic modelling or properties of U(IV)-bearing solutions from the laboratory, there are some of those available in the literature. and perhaps also compare to measurement of U-speciation in highly reducing waters at Forsmark, Sweden (Tullborg et al., 2017, https://www.sciencedirect.com/science/article/pii/S187852201630145X) that show that

U is indeed complexed with carbonate in these solution, but is still U(VI) dominated uranyl-complexes. So the explanation presented here may be a bit oversimplified without further evidence.

Line 267; "...this sample could be a standard U/Pb dating, Sr isotopes, and synchrotron U spectroscopy" this sample looks promising as Sr isotope standard. But a few more samples than 2 (although with two aliquots each?) would be needed to confirm this further.

Summary: line 272: "Details such as the U oxidation state..." Here, and elsewhere, much emphasis is put on the importance of knowing the U oxidation state of the carbonate for U/Pb dating. How exactly is this affecting the geochronology considerations and outcome? I may have missed it in the MS. please add some discussion about this particular matter

---

## Referee Comment (RC2) · Donald Davis (Referee) · 8 Sep 2020

This manuscript describes various imaging methods used to characterize a number of carbonate standards useful for U-Pb geochronology. Carbonate dating is a relatively new and very promising field so the manuscript is useful to geochronologists and will be of great interest to those working in the field. I found it to be generally well written and organized, except for a small number of grammatical mistakes that are noted on the attached annotated copy. The only part that I have a comment on is the suggestion for more use of fission track mapping to outline high-U zones for laser ablation. Neutron activation produces a wide variety of radioactive atoms with varying half-lives so the specimen will remain somewhat radioactive for a long time. Dating carbonate samples requires ablating relatively large amounts of sample, which could

result in making the quadrupole separator somewhat radioactive. This in turn could increase the background of the electron multiplier detector and it is also important to have low backgrounds for this kind of work. Although regular cleaning could minimize this I don't think that I would like to try it with my instrument. Radiography sounds safer.

Please also note the supplement to this comment:
https://gchron.copernicus.org/preprints/gchron-2020-20/gchron-2020-20-RC2-supplement.pdf

**Supplement:**

[revised manuscript text omitted]

---

## Author Comment (AC1) · 29 Sep 2020

We agree with the reviewer that it is unwise to use a sample that has been sent to a nuclear reactor for laser ablation. Instead we would suggest using a companion slab to that.

---

## Author Response (AR1)

Reviewer 1

The manuscript of Rasbury et al describes a toolkit for U-Pb geochronology and specif-ically presents characterization of a couple of used or potentially applicable reference materials. The methods are up to date, although some commonly used characteriza-tion methods for natural samples are missing. The discussion about U speciation is interesting but requires some more connection to published findings of high U incor-poration of U into calcite. I think this manuscript can be published after some minor revisions. Specifically important is the measurements of U(IV) in the calcites, but some more discussion on how the U speciation affect geochronology considerations and re-sults could be suitable. A minor setback is that discussions of processes of U speciation and incorporation and suitability of and heterogeneity of different reference materials are mixed and make the paper a bit un-distinct in its focus.

To our knowledge, the manuscript has referred to all of the published work on the oxidation state of U in carbonates. There is really no evidence that it matters for dating if the U is reduced or oxidized. Rather, the question is how is uranium available to be incorporated. For the three examples we present, we suggested how the uranium was available. To address this concern though, we state up front now that there is no evidence that the oxidation state of uranium matters to the reliability for U-Pb dating.

We added: 'Understanding U incorporation in carbonates is important for a holistic approach to U-Pb dating, but there is no evidence that the oxidation state of U in carbonates has any control on the reliability of the dating. However, it could indicate something about when during diagenesis the U was incorporated, and therefore affect the interpretation of the U/Pb date'

Specific c omments: T itle: I t w ould b e g ood i f t he t itle s pecifies th at it is ca lcite and dolomite that is in focus of this manuscript as there are many other minerals under the "carbonate" umbrella. Title and throughout the MS: consider changing "standards" to "reference materials".

While it is true that we only discussed calcite and dolomite in this contribution, we would argue that the tools are also relevant to other carbonates, and would rather the title be broader so that researchers that are analyzing any carbonate, particularly oneS that they would like to date, would find our paper.

We agree that the use of reference materials is more accurate than standards and this has been globally replaced.

Abstract: It would be helpful if it was stated whether the "split stream" analysis of 87Sr/86Sr together with U/Pb, is done with LA-ICP in the same spot, in nearby spots, or if the 87Sr/86Sr is done on dissolved samples and U/Pb with spot analysis. And, if it is the latter, the later discussion should discuss how the difference in scale of larger dissolved aliquotes match spots, when taking heterogeneity of the material into account.

We have not done split stream analyses. The beauty of split stream though, is that it can be done on the same spots. Our 87Sr/86Sr results are from Thermal Ionization Mass Spectroscopy and the range of values obtained is included to make these possible reference materials available for split stream for those labs that are set up for it.

We added 'While not part of the current contribution, this combination could streamline split stream analyses of 87Sr/86Sr and U/Pb geochronology'. to clarify this to the reader.

Line 20-21: "Mixing of fluids can be particularly corrosive or can be responsible for mineral precipitation." please give more details.

Lines 21 and onwards: "Introduction of C2fluids with different chemistries through uplift or burial can destroy the original carbonate prior to precipitation of a new carbonate

These two comments are coupled and are addressed together here. We added to this statement to clarify, and we added important references for papers that are foundational to this statement.

Mixing of fluids can be particularly corrosive such as seen where seawater and fresh water mix in a carbonate platform, or can be responsible for mineral precipitation depending on the saturation state of the combined fluids \citep{runnells_diagenesis_1969, wigley_mixing_1976}.

Lines 55-58: "Laser ablation mapping (Woodhead et al., 2010, 2007; Piccione et al., 2019; Drost et al., 2018; Roberts et al., 2020), synchrotron XRF mapping (Cole et al., 2004; Piccione et al., 2019; Frisia et al., 2008; Vanghi et al., 2019), PIXE (Ortega et al., 2005, 2003) and µXRF (de Winter et al., 2017) are all ways to take the physical observations of different phases to a new level with in situ chemistry, that maps U in particular, along with other elements that provide details of the fluid chemistry." It would also be good to add the FE-EMPA for spatial characterization of presence of U nano-particles in calcite (see Suzuki et al., 2016 SciRep).

Thanks. We added this reference in the context suggested here.

Because the uranium in this contribution is not in the calcite lattice, but rather U mineral nano-inclusions we added this as well: 'In some cases, uranium minerals are encapsulated in calcite which can protect them from alteration \citet{suzuki_formation_2016,ludwig_uranium-daughter_1978}.'

Lines 61-65: "Given that the major element composition of seawater has changed appreciably through time (Hardie, 1996; Horita et al., 2002), that meteoric fluid com-positions are controlled by water rock interaction (Chung and Swart, 1990) and that deep brines have evolved since deposition in basins (Musgrove and Banner, 1993), we can imagine that there is no such thing as a typical fluid. A holistic approach to dating carbonates should involve an effort to see back to the fluid or fluids that have been responsible for its formation and how they might have changed through the diagenetic history."

At some point in the introduction or in the other text, that surch for growth zonations using SEM, CL, and SIMS C- and O-isotope analyses prior to dating. see Milodowski et al APGEO https://www.sciencedirect.com/science/article/pii/S0883292718301938, and Drake et al NatComm 2019 https://www.nature.com/articles/s41467-019-12728-y. What also could be included in a toolbox for complex samples are fluid inclusion studies that can tell about mixing at particular events and about different fluid flow events

by giving T and salinity estimates. Another tool is in situ Sr isotope values of different growth zones of calcite crystals. see Drake et al., 2019 for the latter.

We agree that all of these tools are important for a full characterization of carbonates and clumped isotopes are useful as well. Many techniques are applicable to characterize the cement stratigraphy of samples. We rephrased our introductory paragraph on techniques to read as follows:

"Often, combining microanalytical tools that characterize carbonates at the micron to centimeter scale of growth zones is required to understand the origin and timing of various textures in complex samples.  Chemical imaging techniques, carbonate staining, and polarized light, fluorescence, electron, and cathodoluminescence microscopy provide spatial context for quantitative microscale analyses of fluid inclusion, ion microprobe, electron microprobe, and laser ablation measurements. Integrated data from such a toolkit enables collection of LA-U/Pb isochrons that reliably date a single paragenetic event and permits geologically meaningful interpretation of U/Pb ages."

Line 129-130: "The cements have mostly been altered to calcite, though Chafetz et al.(2008) found original aragonite in similar cements. Calcite with the greatest alteration is light brown..." This is a bit unclear, is the light brown calcite the most altered of the calcites, or is it replacing aragonite to a larger degree than other calcite types does?

It is almost all calcite as stated : The Walnut Canyon sample has incredible preservation of the original fibrous aragonite texture even though it is almost entirely converted to calcite.  We added:

"While optical orientation is preserved and fluid inclusions and aragonite relict mineral inclusions delineate the original acicular growth habit, the carbonate is almost entirely replaced by cloudy brown calcite (Mazzullo 1980)."

Mazzullo, S. J. "Calcite Pseudospar Replacive of Marine Acicular Aragonite, and Implications for Aragonite Cement Diagenesis." *Journal of Sedimentary Research* 50, no. 2 (June 1, 1980): 409–22. https://doi.org/10.1306/212f7a18-2b24-11d7-8648000102c1865d.

Line 131: "WC-1", mention "Walnut Canyon" at first mention of this material.

Done

Line 135: Fig 3: quality of figure is not publishable. Maybe this has to do with compres-sion for pdf conversion. The veins are difficult to see.

Rescanned and grouped with former Fig. 4

Lines 139-140: "Cathodoluminescence (CL) has been a go-to test for diagenesis, and when an activator such as Mn is present, this often gives phenomenal images that illu-minate alteration." and is also very good for veins and single crystals in open fractures, to track different events of precipitation and/or reacitvation and fluid flow fluctuations.

Agreed, and added. Thank you.

For

Line 144: Fig. 4, 5, 6: These are nice figures, some suggestions; 1. however, what is the difference between Fig 4a, and 5a Since fig 4b and "5-Sr" are the same, maybe 4 can be deleted and 4a inserted (if it is not the same as 5-"photo" but in B&W, the caption does not say.) 2. What is roughly the range of Sr concentrations in these figures (4-5)? It would be very interesting to know as there is also clear drop in Ca in the Sr-rich parts, so I assume Sr concentrations are quite high.

Figure 4a (top) is a CL image of the slab shown in Figure 3. The only overlap in images is that 5Sr is indeed the same as the one we paired with the CL in figure 4. We think it is nice to have it in both places because it is easier to see how it relates to other elements when they are side by side as in Fig 5, but a major point is that the Sr map does a better job of showing alteration than the CL so having them paired together in 4 and blown up to make it easier to convey that seems justified. We are suggesting that as a reference material, we think that maps of each slab and only using areas that are the least altered (which is still more than half of the slab) would result in better reproducibility of measurements of this reference material.

Line 160: Fig 7: This figure is nice, but would to readers (such as me) who reads from left to write, be more logic if starting at left (old) towards right (young). How were the aliquots for Sr prepared, micro-drill, or are they in situ analyses using LA-MC-ICP-MS?details can be given here.

It is certainly reasonable to reflect this graph so that older is on the left- so we changed it as suggested. We added 'The aliquots were microsampled, dissolved and purified using Sr spec resin, and run by Thermal Ionization Mass Spectrometry' to the figure caption. We also changed the scales to zoom in a bit and added a line that is the average of 1, 3, 4, 5 and for the 2SD of that average. While 2 isn't included in the average it plots on the lower line. We left out the tiny slabs of the rock because it isn't possible to make them easy to see and they are described.

Line 160: "...with an average of 0.706930(69)", what is "(69)"? a reference with wrong formatting?

We changed this throughout the text to (0.000069). It is a somewhat normal convention to leave the zeros off but since it was confusing to the reviewer it makes sense to be as clear as possible.

Line 165: "...in seawater through the growth of the cements." Here, fluid inclusions may be useful to give more information on the history of fluids and alteration.

We agree with this and this reviewer is right on target with these comments. However, we start the sentence that without more work it isn't possible to know, and it seems off target to put too much more emphases on this point.

Line 169: TES: define, both in terms of settings and acronym

We do define this in the early section on synchrotron measurements starting on line 94. However, we had not done a good job of talking about the settings so we added a section about the energy range this represents. It now reads:

'These brighter sources offer micron sized resolution for low ppm levels of the elements of interest. Emerging tender energy spectroscopy (TES), using the energy range between soft and hard Xray techniques allows mapping of elements such as Mg and S that are important in carbonates \citep{northrup_tes_2019}.'

Line 173: fig 8, add to caption which sample it is

Done- it is Walnut Canyon and we state that it is the same slab shown in Fig. 3-6.

Line 179-180: "Through neomorphism to calcite, U was left behind because active functional groups such as carboxyl have a high affinity for uranyl." So was uranyl com- plexed to carboxyl in the primary fluid and incorporated into aragonite as carboxyl-complexes. Or did this complexation occur after recrystallization? if the former, could the aragonites have had exceptional U/Sr already at the aragonite stage?

This is a good question. Aragonite doesn't exclude U, but we don't know the U concentration or U/Ca ratio of Permian seawater so we can't know this. There is certainly abundant organic matter and whether the active functional groups represent the degradation products of the original organisms that produced them, or if these were organic molecules that were structurally incorporated is not possible to know. As written, we imply that the U was incorporated in the aragonite and then retained because of this affinity for the organic acid functional groups. We didn't change anything, but if we missed the point of the question we will be happy to revisit it.

Line 189-191; this sentence is not comprehensive, something is missing. and full reference should not be within ().

We fixed the parenthesis and we reworded the sentences relevant to this comment to make it clear we are characterizing a new sample that is large enough to share with the community.

'The sample that we are using to illustrate these tools for characterization is large enough to be suitable for distribution as a secondary reference material for LA carbonate dating (Fig. \ref{fig:BarstowTufa}). Most of the samples used in the \citet{cole_using_2005} study were small slabs from Vicki Pedone (Cal State Northridge) and are too limited in quantity to distribute to the community. Here, we are characterizing a new sample that is large enough to distribute widely.'

Lines 192-193: "Isotope dilution on this sample is under way and will be completed when Covid quarantine is lifted." Such a phrase makes me question if this something that will be included in this MS at revision, otherwise this level of details is probably not adding anything.

AGREE WE WILL EITHER FINISH THIS DATA OR DELETE THE STATEMENT AT SUBMISSION OF THE MANUSCRIPT.

Lines 198-199: "there are mm to cm scale layers of higher and lower U concentrations (Cole et al., 2004)." what are these concentrations: "higher" and "lower"?

We added approximate concentrations in parentheses. 'citet{becker_cyclic_2001} showed that layers in the Barstow tufa deposits have a pattern of low (~10ppm) to high (several hundred ppm) U concentrations across laminae revealed by fission track mapping.'

Line 207: "The 87Sr/86Sr isotope ratios are similar throughout the sample..." this is a bit unspecific. maybe say "show relatively small variability" because there is indeed some variability

For

This is true and we have changed the wording to be more accurate.

'The 87Sr/86Sr isotope ratios of 3 aliquots ranged from 0.719877 to 0.721038 (Table \ref{tab:BT_Sr})'.

Line 209; "...making it a good Sr isotope standard..." I am not certain that the three analyses presented here are enough to state it is a suitable standard for Sr isotope analysis. It is stated that zones with high Sr would narrow the range (also for Sr isotope variability?), I cannot see how this is justified with the data presented, especially since there is no information on Sr-concentrations for table 1.

We agree. This was poorly worded and not well justified. This now reads: 'More work will be needed to determine if this sample can be a Sr isotope reference material as well as a secondary U/Pb carbonate reference material.'

Lines 214-215: "We hypothesize that U(IV) is complexed with some oxyanion in the lake water (phosphate, bicarbonate, etc) that keeps it in solution and perhaps is also incorporated into the calcite lattice." here it would be good to show some thermody-namic modelling or properties of U(IV)-bearing solutions from the laboratory, there are some of those available in the literature. and perhaps also compare to measurement of U-speciation in highly reducing waters at Forsmark, Sweden (Tullborg et al., 2017, https://www.sciencedirect.com/science/article/pii/S187852201630145X) that show that U is indeed complexed with carbonate in these solution, but is still U(VI) dominated uranyl-complexes. So the explanation presented here may be a bit oversimplified with-out further evidence.

The question of U complexation in a fluid and how it becomes part of the carbonate are unquestionably important and remain to be answered. We added a reference to Langmuir because that is foundational work on the fluid side of things and we included this Tullborg ref as an example of a field situation with highly reduced waters and U is still in solution as U(VI). The question then becomes how that is incorporated into the carbonate as U(IV) and this remains to be answered. Thus we scaled back on our interpretations and offered some suggestions for the way forward.

'While it is easy to conceive of a stratified lake with reducing bottom waters, reduced U is insoluble in most solutions, begging the question of how it is available to go into the calcite \citep{langmuir_uranium_1978}. Elevated actinide concentrations are found in the Great Basin lakes, and it is thought that the carbonate alkalinity is responsible for this elevation \citep{anderson_elevated_1982, simpson_radionuclides_1982}. Similarly, \citet{tullborg_occurrences_2017} demonstrates U(VI) complexes in extremely reducing fluids of deep groundwaters in Sweden. It seems clear that there is a kinetic barrier to U reduction even in the most reducing fluids, and yet some carbonates have entirely reduced U in their structures \citep{sturchio_tetravalent_1998}. More work is needed to understand the incorporation mechanism and use of synchrotron facilities to image the distribution of U with respect to other elements, and XAS measurements of U and other redox sensitive elements are important tools for advancing this research. Additionally lab directed studies that can produce U(IV) in carbonates would be a huge leap forward.'

For

Line 267; "...this sample could be a standard U/Pb dating, Sr isotopes, and synchrotron U spectroscopy" this sample looks promising as Sr isotope standard. But a few more samples than 2 (although with two aliquots each?) would be needed to confirm this further.

There are three aliquots, but we agree, we are presenting it as a possibility to be worked on by a number of labs before it could possibly be put out there as a reference material.  This now reads:

87Sr/86Sr  of three aliquots of the Turkana dolomite from the outside (oldest) to the inside (youngest) are indistinguishable from each other at 0.703306 (Table \ref{tab:TBW_Sr}). With the well behaved U/Pb systematics, high concentrations of reduced U and homogeneous 87Sr/86Sr, and further work in collaboration with other labs, this sample could be a reference material for U/Pb dating, Sr isotopes, and synchrotron U spectroscopy.

Summary: line 272: "Details such as the U oxidation state..." Here, and elsewhere, much emphasis is put on the importance of knowing the U oxidation state of the car-bonate for U/Pb dating. How exactly is this affecting the geochronology considerations and outcome? I may have missed it in the MS. please add some discussion about this particular matter

We addressed this in the first instance by saying that there is no indication that the oxidation state of U matters for the reliability of dating.

Associate Editor Comments

**Associate Editor Decision: Publish subject to minor revisions (further review by editor)** (16 Nov 2020)
by Nick Roberts
Comments to the Author:
Thank you for trying to address the concerns and comments of the reviewers. Once these have been made in a final version, the manuscript should be ready for publication. Please decide on the final outcome of statements such as further data after Covid restrictions, and make sure editorial things like "??" are removed.

I only have one real concern over the manuscript, and that is how the data and discussion are portrayed.
I find the title slightly misleading, or at least, the paper does not give what I would expect based on this.
i.e. the paper describes a selection of techniques, but these are not ever going to be the most common by any stretch. CL followed by LA trace element mapping will probably become the most common (CL already is).
Therefore this paper does not provide a comprehensive review of sample characterization tools – I know it does not suggest that it should, but I find that the title sort of suggests that.
The title also does not mention the use of case studies to demonstrate the methods.
The mapping Uranium section describes four methods – you could even describe these as four uncommon methods!
Rather than fully reviewing any of the characterization methods, the paper presents three case studies. I would prefer to see a fuller review of these methods with a wider range of case studies, or, given the short length of the paper, that it be recast (i.e. not involving much, but a change in title and some sentences in the intro and summary) as a demonstration of some common (CL) and uncommon (synchotron-based) methods.

For

We agree with AE Nick Roberts that the manuscript does not review more traditional carbonate petrography. We changed the title to reflect that this manuscript presents case studies based on characterization of U and other elements which builds on traditional approaches.
* * *

[revised manuscript text omitted]

For

[Figure]

Figure 2019. Relative intensities of emission regions for each element detected in Turkana Dolomite. Each panel is gamma and contrast adjusted to emphasize gradients between textures. (a) No incident radiation filter, imaged at 25 µm pixels with 120 ms/pixel dwell time and 1.5 s.d. gaussian blur, with box indicating region re-mapped with filter shown in panel b. (b) Incident radiation filtered using 100 µm Al, 50 µm Ti, 20 µm Cu foil, imaged at 20 µm pixels with 900 ms/pixel dwell time and 1.5 s.d. gaussian blur.

PbTotal_ppm  Mg24_ppm  Si28_ppm  Ca43_ppm

Mn55_ppm  Fe57_ppm  Sr88_ppm  La139_ppm

Nd146_ppm  Eu153_ppm  Dy163_ppm  Er166_ppm

Lu175_ppm  Pb206_ppm  Pb207_ppm  Pb208_ppm

Th232_ppm  U238_ppm  Final U238/Pb206  Final Pb207/Pb206

[Figure]

Figure 2120. Laser Ablation ICPMS maps of the TBW dolomite. U is high concentration, while Pb and Th are low. The REE are also low abundance. The U/Pb ratio is very favorable.

[Figure]

Figure 2221. Map of U. Regions selected based on U concentration. $^{206}Pb/^{238}U$ ages of each region shown on a probability plot.

[Figure]

[Figure]

Table 1. $^{87}$Sr/$^{86}$Sr from aliquots of the Barstow tufa run by Thermal Ionization Mass Spectrometry

| sample | $^{87}$$^{86}$Sr/ Sr | 2SD |
|---|---|---|
| BT1 | 0.720423 | 0.000011 |
| BT2 | 0.721038 | 0.000011 |
| BT3 | 0.719877 | 0.000011 |

Table 2. $^{87}$Sr/$^{86}$Sr from aliquots of the Turkana dolomite run by Thermal Ionization Mass Spectrometry

| sample | $^{87}$$^{86}$Sr/ Sr | 2SD |
|---|---|---|

| TBWi-1 | 0.703309 | 0.000011 |
| TBWi-2 | 0.703299 | 0.000011 |
| TBWo-1 | 0.703306 | 0.000011 |
| TBWo-2 | 0.703312 | 0.000011 |

**For**
**For**
**For**
**For**
**For**
**For**
**For**
**For**
**For**
**For**
**For**
**For**
**For**
**For**
**For**
**For**

**For**

---

## Author Response (AR2)

Dear Troy

you paper is now acceptable for publication in GEOCHRONOLOGY.

However, I have a few requests for technical improvements. Particularly some of the figures are quite poor and could be improved.

Would it be better to use U(superscript)4+ rather than U(IV) , the latter is used to describe coordination of U not charge.

l 292: this sentence is a bit strange. It is fine for a proposal but not a paper.

I deleted this sentence.

Fig 1

why are some terms written with capital first letter (e.g. Rhombs) and other with lower case (e.g. botryoids)?

I changed all to have a capital first letter.

Fig 5 scale invisible

Scale added.

Fig 6: yellow nearly invisible, chose a prettier color, tick marks are small

Changed yellow to green. Made the ticket marks cross the axis so they would be more visible.

Fig 7 scale invisible

Changed the color of the scale bar and words to white. Added the scale to the other images so that it is easily found.

fig 8 abbreviation for second is s not sec in science is U6 the same as U(VI) and you actually mean U(superscript)6+?

Changed.

Fig 9: out of focus?

I replaced this with a photo of the slab that doesn't show the outside of the slab.

Fig 10, very fuzzy, text is almost impossible to read, can you supply a high-res image?

I really want to include this legacy data from the NSLS (never published and really quite a nice dataset) but I am not able to re-process the data- the software needs rates and pixel sizes and I do not have those details. I have cleaned up the text. This figure can be dropped if this isn't good enough.

Fig 13 same issues as Fig 8

Changed.

Fig 14, no scale and fuzzy

Added scale and sharpened slab image.

Fig 15. not acceptable, too fuzzy, text impossible to read, no scale

I changed the color scale to spectrum and instead of exporting the overview, I took the individual maps and changed the fonts to be readable. I left out the Mg and Sr and Mg/Ca, Sr/Ca since the main point is the U/Pb and this allowed me to make those maps larger. I also changed this to be element ppm rather than having the mass number after the element name.

Fig 16: totally fuzzy text

I redid the figures and got rid of the second isochron and probability plot. I made the text larger and tried to scale the maps so that the textures were emphasized. I think this version nicely shows that the criteria I selected for the defined the pooled pixels takes data primarily from the sparry calcite layers. I plotted the tails as different colors but they are not included in the age calculation.

Fig 20: Isotopes are written superscript (Nr)Element, 143Nd not Nd143. They are just pronounced Nd143. Do you show the concentrations of the isotopes here or the element. I think the latter in which case the number needs to be removed. You can mention in the analytical section which isotopes were measured, but here the element abundance would be the preferred value, not the isotope abundance.

True, I was just taking the maps straight from Iolite which is a python code that doesn't allow the numbers to come first (I also corrected this in fig. 16) . I also realized that I do not need to export all of these maps. Some show nothing and others are redundant. I have tightened this up and pasted correct labels on the maps that remain.

Fig 21: isotopes are superscripts, fuzzy images

I changed this to make the text more readable. I got rid of maps that didn't show much, I added a scale bar, and I change the palette to spectrum instead of ion so that the colors are cool-warm (low-high).

Fig 22 same issues as Fig 9, Figure caption missing

I fixed these issues.